# Brain Functional Correlates of Resting Hypnosis and Hypnotizability: A Review

**DOI:** 10.3390/brainsci14020115

**Published:** 2024-01-24

**Authors:** Vilfredo De Pascalis

**Affiliations:** 1Department of Psychology, La Sapienza University of Rome, 00185 Rome, Italy; vilfredo.depascalis@uniroma1.it; 2School of Psychology, University of New England, Armidale, NSW 2351, Australia

**Keywords:** functional neuroimaging, EEG oscillations, functional connectivity, hypnosis, hypnotizability, resting, cognitive neuroscience

## Abstract

This comprehensive review delves into the cognitive neuroscience of hypnosis and variations in hypnotizability by examining research employing functional magnetic resonance imaging (fMRI), positron emission tomography (PET), and electroencephalography (EEG) methods. Key focus areas include functional brain imaging correlations in hypnosis, EEG band oscillations as indicators of hypnotic states, alterations in EEG functional connectivity during hypnosis and wakefulness, drawing critical conclusions, and suggesting future research directions. The reviewed functional connectivity findings support the notion that disruptions in the available integration between different components of the executive control network during hypnosis may correspond to altered subjective appraisals of the agency during the hypnotic response, as per dissociated and cold control theories of hypnosis. A promising exploration avenue involves investigating how frontal lobes’ neurochemical and aperiodic components of the EEG activity at waking-rest are linked to individual differences in hypnotizability. Future studies investigating the effects of hypnosis on brain function should prioritize examining distinctive activation patterns across various neural networks.

## 1. Introduction

A peculiar characteristic of the human brain lies in its ability to transform endogenous mental representations into perceptual states. The construction of perceptual states involves a top-down dynamic interplay between sensory processing, memory systems, attentional mechanisms, and higher-order cognitive functions. This complex composition allows us to construct a coherent and meaningful perception of the external world based on our internal mental representations. Such a top-down process is modulated by hypnosis, a social interaction in which, in the classic instance, the participant responds to verbal suggestions for imaginative experiences conveyed by the hypnotist involving distortions in the environment awareness, e.g., conscious perception and memory, and sense of agency [1,2,3,4,5]. Hypnosis is a special rapport between the hypnotized person and the hypnotist in which the subject becomes deeply absorbed and focused on the hypnotist’s voice with disconnection from extraneous stimuli and the letting go of thoughts [2,6]. According to Reyher [7], hypnosis research can be split into “intrinsic” and “instrumental” main approaches. Intrinsic research attempts to understand the phenomena of hypnosis itself and to disclose the components of hypnotic responding.

In contrast, instrumental research uses hypnosis as a device to study a variety of psychological processes, including a dynamic interplay between sensory processing, memory systems, attentional mechanisms, and higher-order cognitive functions (involved, e.g., in functional amnesia, functional visual disorders, paranoia, false memories), to facilitate research in other fields (for details see [8]). By adopting the intrinsic approach, Cardeña and Spiegel [9] suggested three main components for hypnosis: absorption, dissociation, and hypnotic suggestibility. The term absorption refers to the degree to which one participant is intensely focused on a mental experience, while dissociation concerns the mental disengagement from the external environment [9,10,11]. Hypnotic suggestibility (or hypnotizability) refers to the individual ability to respond to hypnotic suggestions involving self-orientation and automaticity, i.e., suggested responses are experienced as being produced involuntarily and effortlessly [6,11,12]. Hypnotizability varies among people, with some individuals being more quickly and deeply hypnotizable while others may have more difficulty responding to suggestions. Experiential hypnotizability, i.e., a measure of experiential involvement in hypnotic suggestions, has been associated with the tendency to experience alterations in consciousness as boundary lessness, self-transcendence, and absorption [13]. Hypnotizability is generally considered stable across the lifespan [14]. Still, it has also been reported that it is not a stable trait and can be modified [15,16]. It varies across a single day [17,18] and between sessions [19]. However, no one has specified the neural processes underlying behavioral responses to hypnotic suggestions, and little research has been focused on the specific factors that optimize responsiveness to hypnotic suggestions [20,21]. These findings raise the question of whether there are differences in brain activity between people who are naturally highly hypnotizable individuals (HHs) and those who are trained HH individuals, which can be a new topic for future research.

Over the past two decades, extensive research into the neural aspects of hypnosis and hypnotic responsiveness has yielded tangible evidence of objective changes in the brain resulting from hypnosis. However, the findings from various studies appear to be in conflict and have sparked controversy, as detailed in reviews by Landry et al. [22] and Vanhaudenhuyse et al. [6]. 

According to Terhune [23], it is improbable that a single mechanism can comprehensively elucidate the entire spectrum of hypnotic phenomena. Hypnosis is a multifaceted process encompassing absorption, embodied relaxation, alterations of self-perception, and changes in agency, and none of these elements individually constitute the entire spectrum of hypnosis phenomena. Neural findings, on the whole, confirmed the complexity of the hypnosis phenomena and indicated that the neural mechanisms of hypnosis and hypnotizability are not fully understood [24]. For instance, even though the relevance of hypnosis as an altered state of consciousness has been questioned and hypnotic suggestions have been consistently and efficaciously used to treat clinical pain [25,26,27], the specific top-down cognitive processes underlying responsiveness to hypnotic suggestions are still poorly explored and explained [5,28]. Barnier and Nash [29] have suggested that the lack of a widely accepted definition of hypnosis is one cause of the limited research on hypnosis and the scarce knowledge of mechanisms characterizing hypnosis. Hypnosis is a complex phenomenon embodying several elements, such as interpersonal interaction, suggestion, relaxation, focused attention, concentration, imagination, mental peace, altered perception of the environment, amnesia, change in emotional perception, disengagement of the discursive and critical analytical reasoning [30]. This complexity makes it difficult to determine whether a specific treatment should be classified as hypnosis. Thus, researchers and clinicians need to specify the definition and reference model of hypnosis they use in their work. Recently, Jensen and colleagues [28] have suggested several critical recommendations for a research agenda for the next decade to solve this question. They mainly underlined as essential (i) the use of data sharing, (j) redirecting resources away from studies comparing state and non-state hypnosis models to neuro-clinical studies evaluating the efficacy of hypnotic treatments in the influence of central nervous system processes, (k) the neurophysiological underpinnings of hypnotic phenomena and individual differences in hypnotizability [31]. 

Several studies have explored the neural underpinnings of hypnosis, utilizing methods such as noninvasive electrophysiology and brain imaging. However, even after extensive exploration, reaching a consensus on the specific brain regions and functions responsible for hypnotic experiences remains challenging [22,32]. Disagreement arises regarding the use of electrophysiological measures in assessing hypnosis. Exploring these markers has yielded inconsistent findings, primarily due to methodological variations among studies. However, amidst this disparity, a few potential trends have emerged. For instance, heightened theta power has been proposed as a potential biomarker for hypnosis [32,33]. Yet, Farahzadi and Kerkecs [34] have criticized the proposal of theta rhythm as a biomarker of hypnosis since it remains uncertain whether theta activity is merely a consequence of the procedural aspects inherent in hypnosis or plays a causal role in inducing hypnotic responsiveness. This uncertainty persists because, during any verbal exchange—whether hypnotic or not—the listener’s auditory cortex synchronizes with the 3–8 Hz rhythm of the speaker’s syllable production rate, thereby enhancing the ongoing theta oscillations (4–8 Hz) in the listener’s auditory cortex [35]. 

Neuroimaging research also delves into the intricacies of modulations within extensive neural networks, particularly the Default Mode Network (DMN), the Salience Network (SN), and the Executive Control Networks (ECN). The focal points of neural exploration of hypnosis have been the nodal regions within these networks, primarily involved in regulating top-down attentional processes. However, neuroimaging correlates of hypnosis/hypnotizability studies, as reviewed by Landry and colleagues, have been proved to be inconsistent [22]. The inconsistencies across these research findings partly stem from the lack of standardized experimental approaches in studying hypnosis. Varying induction techniques, suggestions, and experimental designs across studies significantly impact brain activity and hypnotic responsiveness. Standardizing experimental paradigms could potentially alleviate some of these discrepancies [28,36]. In essence, unraveling the enigma of hypnosis necessitates navigating its intricate web of variables, standardizing methodologies, and acknowledging its multifaceted nature as a complex cognitive stimulus [22,32].

The current review is motivated by recent novel hypnosis/hypnotizability findings from EEG oscillatory and functional neuroimaging research. These add unknown knowledge to the neurophysiological underpinning of resting hypnosis and individual differences in hypnotizability. 

This review is based on all the research/review papers providing reliable statistical information derived from PsycArticles, MEDLINE, Scopus, and Science Citation Index. The investigation was limited to studies conducted on healthy adult human samples without restrictions on gender or ethnicity. The research on electronic database for EEG oscillation studies were limited to the years 1962–2023 and included the following search string and keywords: (hypnosis OR hypnotizability OR hypnotic OR suggestibility OR resting-state OR suggestion OR hypnotic state OR consciousness OR susceptibility OR attention OR mental practice OR cognitive task OR resting state OR intention OR loss of control OR awareness of movements OR autogenetic training OR perception OR paralysis OR inhibition OR emotion OR behaviour OR behavior OR possession tranceOR passivity OR regulation of consciousness OR attention) AND (EEG OR electroencephalography OR fMRI OR functional magnetic resonance imaging OR PET OR positron emission tomography OR SPECT OR single photon emission computed tomography OR CT OR computer tomography OR regional cerebral blood flow OR neuroimaging OR structural and functional cerebral correlate OR functional connectivity OR local neuronal activity OR functional brain activity change OR brain activity OR brain imaging OR mental imagery OR resting-state functional connectivity OR cerebral hemodynamics OR whole-connectivity profile OR voxel-based morphometry). The review allows the selection of two hundred and seventy studies. 

The main goals of this review are to summarize the current state of research findings regarding (1) the associations between hypnotizability/resting-hypnosis and structural and functional neuroimaging measures, (2) EEG oscillation activities as well as EEG functional connectivity correlates of hypnosis and hypnotizability (3) discuss the implications of this knowledge to understand the neural correlates of hypnosis and hypnotizability and potentially enhancing the efficacy of hypnotic treatment. 

To achieve these aims, we first briefly describe the physiologic underpinning of EEG signals, including EEG functional connectivity and functional neuroimaging techniques.

## 2. EEG Spectral Analysis and Functional Neuroimaging Techniques

The EEG and neuroimaging techniques involve registering electrical activity from the scalp using EEG electrodes mounted on a cap and signal processing through pre-amplification and amplification. These data are analyzed in temporal and frequency domains. Time domain analysis yields event-related potentials (ERPs) that reflect cortical activity linked to specific events and provide valuable insights into response timing of millisecond-level changes in cortical electrical activity.

In the frequency domain, the EEG oscillatory activity is quantified by its frequency (i.e., the rate of repetition of the oscillatory event), amplitude or power (i.e., the magnitude of the oscillatory signal at a given frequency), and phase (a measure of its position in time relative to a pre-defined cycle or close to a reference oscillation at the same frequency).

Resting-state oscillatory activity is traditionally defined as the superposition of oscillations falling within distinct frequency bands. The primary frequency bands of normative EEG oscillatory activity in adulthood are named delta (1–4 Hz), theta (4–8 Hz), alpha (8–13 Hz), beta (13–30 Hz), and gamma (>35 Hz). One of the most common signal frequency processing methods is a fast Fourier transform (FFT). An FFT provides the frequency power spectrum for a period, often averaged across a range of frequencies comprising a band (e.g., alpha). It also provides a phase spectrum. The power spectrum reflects the energy of each frequency determined by the squared amplitude of the wave. The phase spectrum reflects the phase in radians or degrees of the sine or cosine wave at each interval. Most frequency analyses focus exclusively on frequency power [37]. 

Relatively to the spectral power measurement, recent research exploring EEG has highlighted a significant factor that might surprise or complicate results in resting EEG. Specifically, the power distribution within EEG frequency bands comprises two key components: a rhythmic, periodic aspect and an irregular, overlapping, aperiodic component. This aperiodic activity showcases power across all frequencies under a 1/f power-law pattern. It reaches its peak at lower frequencies and steadily exponentially decreases as frequency rises [38]. The periodic component is characterized by the oscillatory peak’s bandwidth center frequency and relative amplitude.

In contrast, the aperiodic component is defined by the slope and offset or intercept (i.e., the position where this slope starts) of the power spectrum. The intercept might indicate heightened spiking in neural populations [39,40]. Even if aperiodic activity does not offer meaningful information to the EEG signal, mixing it with periodic activity, as has been the case in the majority of the resting-EEG studies, can introduce noise effects in conventional EEG frequency analyses, as exemplified recently by Ouyang and colleagues [41]. These authors explored if cognitive processing speed correlated with alpha-range activity, as prior studies indicated. After separating periodic and aperiodic components within the alpha band, they found a link between mental processing speed and total alpha power before isolating them. However, this correlation vanished when considering only the actual periodic part of the signal. It persisted solely when the aperiodic portion was analyzed, i.e., measuring the aperiodic slope exponent and offset. As a result, both elements play a role in the overall power computed within distinct frequency bands, such as the alpha waves. This fact implies that the pure rhythmic activity is tied with non-oscillatory power that does not truly oscillate within that specific frequency range. This surprising outcome reshapes the understanding of earlier findings, emphasizing the psychological significance of aperiodic EEG data—challenging its classification as mere “noise.” These observations have highlighted the crucial need to differentiate between these distinct EEG activity patterns in research. Studies on EEG frequency oscillations obtained during several different behavioral tasks have outlined a link between EEG activity and various aspects of brain functions, encompassing sensory processing, perception, motor control, and cognitive activities such as attention, learning, memory, and emotional processing. These oscillations essentially serve as a functional code for the brain and are believed to facilitate communication between different brain regions and support associative processes [42,43,44,45]. 

### 2.1. EEG and Neuroimaging

The interpolation of EEG signals into scalp tomography and source analysis aids in understanding temporal resolution. However, these methods lack precision in spatial resolution, failing to pinpoint the origin of the brain’s activity. This discrepancy, known as the inverse problem, arises because diverse cortical distributions could generate identical scalp activity patterns. Additionally, variations in skull thickness influence how brain activity manifests on the scalp.

While EEG offers insights into underlying brain activity, techniques like dipole modeling simulate different brain generators to match theoretical scalp EEG patterns with actual recordings. However, for more precise localization, PET and MRI imaging techniques surpass EEG’s capabilities (for a detailed comparison, see [46]. Various classification methods exist to distinguish distinct topographies in multichannel EEG data, aiming to describe EEG as potential maps with different time courses [47,48].

Understanding large-scale brain network communication has become pivotal. Analytical methods estimating connectivity in networks, applicable to both fMRI and EEG, have gained traction due to EEG’s high temporal resolution and direct measurement of synchronized neuronal activity across various frequencies. Functional connectivity measures, such as cross-correlation and phase synchronization, are standard, while newer methods assess effective connectivity, capturing causal relationships within brain circuits [49,50,51]. These advanced methods, rooted in Granger’s causality theory, offer insights into brain connectivity. For further details on EEG and brain imaging tools, refer to Michel and Murray’s comprehensive review [47]. 

Among multivariate methods for functional connectivity analysis, those using graph theory are providing appreciable improvement in understanding complex brain functioning. Graph theory provides the means for characterizing the available connections in the brain using a complex network model [52]. This method assumes that the brain is represented as a ‘graph’ composed of ‘nodes’ (brain regions or electrodes) connected by ‘edges’ (links or connections between these regions). Measurement of the strength of connections between nodes usually uses measures like the weighted Phase Lag Index (wPLI) to account for volume conduction effects on brain connectivity estimation [53]. Adjusting the connectivity matrix makes it possible to focus on solid connections without breaking down the examined network. This procedure allows us to assess how the brain is organized and how information flows within it using specific graph theory measures. These most used measures are the ‘normalized clustering coefficient’ (which shows how tightly connected neighboring nodes are) [54], the ‘normalized participation coefficient’ or ‘global efficiency’ (which measures how efficiently information spreads across the entire network) [55], and ‘global modularity’ (which estimates how the network is divided into distinct communities) [56] (Newman, 2006). These measures provide insights into how different brain regions interact within the network and enable the analysis and prediction of human behavior. For instance, using these metrics, some researchers have evaluated how the brain’s organization changes following hypnotic induction and how social factors such as poverty and growth faltering in early experience shape brain networks in children living in low-income countries [57].

### 2.2. PET and fMRI Methods

PET and fMRI methods are fundamental for understanding brain activity. Positron emission tomography (PET) explicitly measures cerebral blood flow (CBF) changes related to brain functions, deriving energy from oxygen and glucose carried through blood flow. Analyzing these flow alterations in various brain regions helps identify active areas during specific tasks. PET involves participants inhaling or being injected with a radioactive isotope, providing the tracer recorded by the PET scanner. The general procedure is to make a measurement, enabled by a gamma ray detector, during a control task, which is subtracted from the measurement taken during an experimental task. While PET’s temporal resolution might be compromised due to the time it takes for readings (minutes), it excels in pinpointing active brain regions during different processing types. Its versatility in measuring radioactively labeled molecules facilitates insights into perfusion, metabolism, and neurotransmitter turnover. However, PET’s drawbacks include cost, reliance on a cyclotron for producing radioactive agents, limited experimental sessions due to tracer injections, and constrained temporal resolution. Moreover, the risks associated with exposure to radioactive tracers limit participants to one study per year, impacting the study of short-term treatment efficacy. For further medical engineering aspects of PET, see [58].

Functional magnetic resonance imaging (fMRI) also relies on blood flow changes in the cortex to identify active areas but uses a different technology from PET. An external magnet in fMRI detects local magnetic fields, analyzing hemoglobin’s magnetic properties pre- and post-oxygen absorption to map cortical blood changes and infer neuronal activity. Ogawa et al.’s groundbreaking research sparked significant enthusiasm for utilizing electrophysiology experiments to elucidate the neural basis of the blood-oxygen-level-dependent (BOLD) fMRI signal in human studies. In 1992, three distinct research groups separately achieved results in humans using the BOLD mechanism [59,60,61], initiating the surge of fMRI publications that have consistently emerged in scientific journals since then (for further details, see [62]). While fMRI boasts better spatial resolution and the ability to generate multiple images from a single individual compared to PET, its physical setup might be uncomfortable for some individuals and limits certain studies.

There are advantages to combining fMRI and PET; using fMRI gives us clear structural brain scans, while PET allows us to index blood oxygen-dependent activity throughout the brain. However, while neuroimaging can specify the activated regions during mental states or cognitive tasks, it cannot determine the ‘necessary’ brain areas for the functions of interest. Traditionally, lesion studies offered insights, but purposely inducing lesions in humans is unethical. Transcranial magnetic stimulation (TMS) emerged as an alternative, allowing brief inhibition or facilitation of specific brain areas. TMS applies electromagnetic induction principles, inducing cortical stimulation through electric pulses to the scalp, resulting in neuronal depolarization and cortical activity. TMS has been extensively used in cognitive neuroscience, inhibiting or stimulating various cortical areas involved in perception, memory, and cognitive paradigms.

Repetitive Transcranial Magnetic Stimulation (rTMS), with frequencies ranging from 0.3 to 20 Hz, modulates cortical activity based on stimulation rates. Slow rTMS (≤1 Hz) inhibits, while fast rTMS (>1 Hz) facilitates cortical activity, providing a means for selective brain modulation in cognitive studies (for a review, see [63]).

## 3. Functional Neuroimaging Correlates of Resting Following Hypnotic Induction without Specific Suggestions

Several neuroimaging studies have demonstrated that hypnotic induction without task-specific or indirect suggestions induces plastic changes in neuronal activity by mainly engaging the frontal and thalamic areas (e.g., [22,30,64,65,66,67,68,69,70,71,72]). Figure 1 depicts the main neural networks, and the associated cortical brain regions found sensitive to hypnosis modulations and individual differences in hypnotizability in the current review. Above all, several reports indicate that hypnotic induction tunes higher-order neural systems involved in higher-order cognitive functions, supporting the top-down view of hypnosis. These observations are consistent across various experimental conditions, including the Stroop task, mental imagery, administration of noxious stimulations, as well as in resting state and neurophenomenological studies [9,64,65,70,73,74,75,76,77,78]. Dissociated control and cold control theories have resulted in being the most used models to explain experimental hypnosis data. These theoretical models served to design experiments well and formulate testable hypotheses (e.g., [79,80]). The Dissociated Control Theory proposes that hypnosis can be explained by a breakdown in communication between executive and monitoring systems [81,82]. In contrast, the Cold Control Theory suggests that hypnosis primarily involves unconscious executive control [83]. According to this perspective, hypnotic experiences arise from the interference of meta-representations in the selection and execution of responses, resulting in incorrect higher-order assessments of intentions for thoughts and actions [79].

Research on the neural underpinnings of agency connects executive control to the retrospective monitoring of the fluency between intentions and actions and the resultant sense of control [84,85]. While the executive system effectively responds to suggestions, hypnotized individuals’ perception of a lack of ownership over their thoughts and actions is attributed to inaccurate metacognitive representations [86]. However, it is essential to note that these theories often settle on the traditional suggestion effect (expressly, the disruption of agency) and, therefore, may have limited explanatory scope. 

In a PET study, Maquet and collaborators [69] (Table 1) have shown that subjects, during revivification of pleasant autobiographical memories in hypnosis, experienced significant activations of a widespread, mainly left-sided, set of cortical areas involving occipital, parietal, precentral, premotor, and ventrolateral prefrontal cortices and a few right-sided regions (occipital and ACC). Rainville et al. [70] (Table 1) used PET to highlight that hypnotic relaxation increased the involvement of the anterior cingulate cortex (ACC), thalamus, and the ponto-mesencephalic brainstem. Hypnotic relaxation further involved increased occipital regional cerebral blood flow (rCBF), suggesting that hypnotic states are characterized by decreased cortical arousal and a reduction in cross-modality suppression (disinhibition). In contrast, increases in mental absorption during hypnosis were associated with rCBF increases in a distributed network of cortical and subcortical structures previously described as the brain’s attentional system. Egner et al. [64] used event-related fMRI and EEG coherence measures to link individual differences in hypnotizability to the efficiency of the frontal attention system. During Stroop task performance in hypnosis, they found a functional dissociation between conflict monitoring and cognitive control processes. Hallsband [30], using a PET scan, investigated the neural mechanisms of encoding and retrieval of high-imagery words in HHs under hypnosis and in waking state. Encoding under hypnosis was associated with more pronounced bilateral activations in the occipital cortex and the prefrontal areas compared to learning in the waking state. Word-pair retrieval, learned under hypnosis, produced activations in the occipital lobe and cerebellum. Perceptual changes can also occur during hypnosis. Kosslyn et al. [68] conducted a PET study demonstrating that color areas in the brain were activated under hypnosis when subjects were asked to perceive color, whether or not they were shown the color or a grey-scale stimulus. These findings highlight how changes in subjective experience during hypnosis can be reflected in brain function among highly hypnotizable subjects. Later, Halsband and collaborators [2,87], using fMRI and Granger causality mapping, confirmed previous findings that hypnosis induces activation in color processing areas when subjects are suggested to perceive color while viewing grayscale stimuli and indicate that visual illusion in hypnosis produces changes in effective connectivity among fusiform gyrus, anterior cingulate cortex, and parietal areas.

Studies that investigated functional neuroimaging correlate during rest following hypnotic induction showed increased brain activity in the anterior part of the default mode network (DMN) during rest in HH individuals. The DMN comprises active brain regions without goal-directed activity (Figure 1 and Table 1). It involves the posterior cingulate cortex (PCC) and preCu (PreCu), medial prefrontal cortex (mPFC), pregenual cingulate cortex, temporoparietal regions, and medial temporal lobes. It is implicated in episodic memory retrieval, self-reflection, mental imagery, and stream-of-consciousness processing [88,89,90]. In the McGeown et al. study [9] (Table 1), the HHs, compared to LHs, showed a more pronounced selective reduction in resting state activity in the medial prefrontal cortex (i.e., the anterior parts of the default mode circuit). 

Suggestions delivered within the context of hypnosis can induce dynamic changes in brain activity [24,32,91,92]. It has been observed that during hypnosis, successful suggestions to modify the sensory and emotional components of a given stimulus can enhance activity in the brain regions in which the modulation of functional connectivity between the ACC and the involved different brain regions is relevant [2,93]. For example, the effectiveness of hypnosis in reducing pain sensation has been well documented [65,70,71,93,94], and the anterior midcingulate cortex has been suggested as the brain region mediating hypnosis-induced analgesia. These hypnosis analgesia findings indicate that pain reduction under hypnosis involves enhanced functional connectivities between the midcingulate cortex and brain regions as insular, pregenual, frontal, and pre-supplementary motor area as well as the brainstem, thalamus, and basal ganglia. These findings align with those reported in a recent pain study [64], indicating that the midcingulate and insular cortex may serve as a network hub that integrates information to create pain perception and contribute significantly to its modulation.

**Table 1 brainsci-14-00115-t001:** Experiments that investigated functional neuroimaging correlates of resting state following hypnotic induction.

Study	Sample	Experimental Task	Technique	Main Results
**Structural Neuroimaging**				
Rainville et al. (2019) [78]	HHs, HHs and LHs (N = 28, 14 men)	Resting state following neutral hypnotic induction	MRI, arterial spin labeling scan (ASL)	An increase in perception of automaticity was associated with enhanced activity in the parietal operculum and in the anterior-supra callosal mid-cingulate cortex (aMCC)
**Functional l Neuroimaging—Resting Condition**				
Deeley et al. (2012) [73]	HHs (N = 8, 4 women)	Resting state following hypnotic induction	fMRI	Self-Rated Hypnotic Depth: Positively correlated with ECN (right MFG, bilateral IFG, bilateral PreCG) and negatively with DMN (left MFG, right ACC, and bilateral PCC; and bilateral PHGs)
Demertzi et al. (2011) [74]	High Dissociation and Absorption scores (level > 6/10; N = 12, 4 women)	Resting state following hypnotic induction	fMRI	(1) Increased anterior DMN connectivity, whereas its posterior midline and PHG structures decreased their connectivity; (2) reduced “extrinsic” lateral frontoparietal connectivity
Lipari et al. (2012) [95]	N = 1 Women hypnotic virtuoso	Resting state following hypnotic induction	fMRI and EEG	Enhancement of activity in posterior regions of the DMN (PreCu, PostCG, retrosplenial cortex, IPL, and PH) and decreased activity in anterior DMN areas (mPFC, MFG, ACC)
Maquet et al. (1999) [69]	N = 15 HHs (N = 11 women)	Resting state following hypnotic induction	PET	Activation of left-sided occipital, parietal, precentral, premotor, and ventrolateral prefrontal cortices and a few right-sided regions, including occipital and anterior cingulate cortices.
McGeown et al. (2015) [96]	HHs (N = 7), MHS (N = 9) and LHs (N = 13); (N = 29, 12 women)	Resting state following hypnotic induction	fMRI	Less connectivity in the anterior part of the DMN
McGeown et al. (2009) [9]	HHs (N = 11) and LHs (N = 7)	Resting state following hypnotic induction	fMRI	There is less activity in the anterior part of the SN and DMN
Rainville et al. (1999a,b) [71,78]	HHs only (N = 8, 3 women)	Prehypnosis-resting state, hypnosis relaxation, and hypnosis relaxation plus suggestions to alter pain unpleasantness	PET and EEG	Modulation of CEN, SN, and DMN: Hypnosis-relaxation-related rCBF decreases in the right IPL, left PreCu, and PCC. Hypnosis-with-suggestions-related increases in rCBF in the left frontal cortices and the medial and lateral posterior parietal cortices
Rainville et al. (2002) [70]	N = 10 HHs only (N = 6 women)	Resting state following hypnotic induction	PET	Modulation of CEN, SN and DMN: Involvement of the ACC, thalamus, and ponto-mesencephalic brainstem and increase in occipital rCBF in hypnosis relaxation; Enhanced rCBF with increased mental absorption during hypnosis in a distributed network of cortical and subcortical structures known as the brain’s attentional system
Jiang et al. (2017) [67]	HHs (N = 36) and LHs (N = 21)	Resting state following hypnotic induction	fMRI	Enhanced functional connectivity between the DLPFC of the ECN and insula in the SN, and reduced connectivity between the DLPFC and PCC in the DMN (PCC)
Vázquez et al. (2023) [97]	MHs and HHs (N = 24, 12 women)	Resting state following neutral hypnosis induction	fMRI	Higher connectivity in the neutral hypnosis than in the resting condition for the frontoparietal cortices of the dorsolateral attention network (DAN), SN, and SMN. Parietal and occipital regions displayed increased network connectivity, implying dissociation from the frontal cortices
de Matos et al. (2023) [98]	HHs (N = 55, 37 women)	Two different depth levels of neutral hypnosis and respective control states	fMRI	Hypnosis: (1) whole-brain analysis disclosed key neural hubs in parieto-occipital-temporal areas, cuneal/precuneal and occipital cortices, lingual gyri, and the occipital pole; (2) Comparing both hypnotic states directly revealed depth-dependent connectivity changes, notably in left superior temporal/supramarginal gyri, cuneus, planum temporale, and LGs

Abbreviations: HHs, high hypnotizable subjects; LHs, low hypnotizable subjects; MRI, magnetic resonance imaging; fMRI, functional MRI; PET, positron emission tomography; rCBF, regional cerebral blood flow; EEG, electroencephalogram; aMCC, anterior mid-cingulate cortex; DMN, default mode network; SN, salience network; ECN, executive control network; SMN, sensorimotor network; DAN, dorsolateral attention network; ACC, anterior cingulate cortex; PCC, posterior cingulate cortex; PH, parahippocampus; PHGs, parahippocampal gyri; MFG, middle frontal gyrus; IFG, inferior frontal gyrus; IPL, inferior parietal lobule; PreCu, precuneus; PostCG, post-central gyrus; mPFC, medial prefrontal cortex; DLPFC, dorsolateral prefrontal cortex; LGs, lingual gyri.

Furthermore, in a more recent study, Rainville and colleagues [78] (Table 1) assessed resting-state brain activity before and after a ‘neutral hypnotic induction’ using arterial spin labeling (ASL, a functional magnetic resonance imaging method sensitive to brain perfusion [99] (Table 1). The study disclosed a positive association of enhanced perceived self-reported automaticity during resting-state in hypnosis with the activity in the parietal operculum and the anterior part of the subcallosal anterior mid-cingulate cortex (aMCC). These findings are in line with previously reported findings that increases in self-reported automaticity at hypnotic rest (i.e., without any stimulus or task) are positively associated with the activity in the parietal operculum [100] and with the frontoparietal network involved in phenomenological aspects of self-agency and volition [101] and demonstrate that these effects can be evidenced at rest, in the absence of overt motor challenges. In addition, these findings indicate that hypnotic induction alters brain regions’ functioning in phenomenological aspects of self-agency and volition in a way that self-produced actions are experienced as being external and nonvolitional. These results have implications for understanding the brain mechanisms underlying delusions of alien control, i.e., hallucinatory symptoms associated with schizophrenia in which patients misattribute self-generated actions to an external source [102,103].

In a comprehensive review of several fMRI study outcomes, Landry and co-workers [22] have offered a framework for exploring the neural correlates of hypnotic phenomena. They brought attention to the SN, ECN (both identified as task-related networks), and DMN (a network active during rest periods) as pivotal networks involved in hypnotizability, hypnotic induction, and responses to hypnotic suggestions (Figure 1). The DMN was predicted to deactivate during the hypnotic induction, while ECN and SN were expected to exhibit heightened regional activities. In addition, subsequently to the induction phase, it was anticipated that HHs would experience increased intra-network interactions within SN and ECN. However, their in-depth examination of study findings, utilizing the Activation Likelihood Estimation method (ALE), contradicted this perspective. Their results indicated that, except for the lingual gyrus, other brain regions did not consistently display patterns of activation or deactivation across studies [22].

The SN encompasses regions such as the dACC, frontoinsular cortices, preCu, posterior cingulate cortex (PCC), anterior insula (Figure 1), amygdala, and ventral striatum. Its primary function is to detect, integrate, and filter relevant interoceptive, autonomic, and emotional information [90,104]. The ECN is central to focusing on relevant information, deploying mental strategies to generate a reliable hypnotic response, anticipating and preparing for the hypnotic response, and assessing the subjective sense of agency. Components of the ECN include the dorsolateral prefrontal cortex (DLPFC) and lateral parietal cortices, which are critical for selecting and retaining in working memory the relevant information needed for preparing actions [90,104]. It is suggested that functions associated with the ECN could contribute to generating and maintaining mental imagery, a valuable aspect of hypnosis. Consequently, this neural pattern is likely to influence the activity of the DMN, which may also be involved in generating mental imagery [105,106]. 

A decrease in activity in the frontal part of the DMN corresponds to a reduction in self-related or internally directed thoughts [9,107]. Changes in ECN activity during hypnosis may also accord to shifts in subjective evaluations of one’s agency during the hypnotic experience, as proposed by the Dissociated and Cold Control theories of hypnosis [108,109]. Research has shown that specific hypnotic suggestions can induce a credible, compelling delusion with features strikingly similar to clinical cases, e.g., mirrored self-misidentification [110]. This research delves into how hypnotic procedures affect belief formation in healthy subjects, demonstrating their ability to replicate certain aspects of monothematic delusions observed in schizophrenia patients [111]. Moreover, it has been observed that being in hypnosis alone can induce several monothematic delusional beliefs, as hypnosis impairs belief evaluation since research has shown that a hypnotic induction diminishes the ability of HHs to differentiate between suggested and actual events [112,113].

The SN plays a crucial role in processing signals from both internal and external sources to regulate how the brain responds to the detection and perception of sensory, interoceptive, and affective stimuli [114,115]. It is important to note that the activity of internal and external awareness networks has been recognized as inversely correlated [116,117,118]. The inner awareness network is primarily activated by stimuli related to oneself, while the external awareness network responds to the stimuli associated with the environment [116,117,118]. The internal awareness network involves midline brain regions such as the medial prefrontal cortex, anterior cingulate, and posterior cingulate cortex. In contrast, the external awareness network mainly engages bilateral frontal and parietal areas. The SN is activated when individuals experience hypnotic absorption and focus on specific mental experiences [11,79]; its activity may reflect the monitoring of attentional focus, prioritizing relevant signals while filtering out irrelevant ones from conscious awareness [119]. Consistent with this view, one study found that the self-reported level of hypnotic absorption correlated with the activity in two central nodes of the SN, the anterior insula and cingulate cortices [70].

Furthermore, fMRI research [96] suggests that the significantly higher grey matter volume (GMV) in the medial frontal cortex and ACC and lower connectivity in the DMN during hypnosis facilitate experiences of greater hypnotic depth. Additionally, more significant GMV in the left temporal-occipital cortex was associated with greater hypnotizability. In another study, an increased connectivity between the DMN and ECN following hypnotic induction was reported [67].

Evidence demonstrates that the SN mediates the ECN and DMN [89,120,121,122]. Changes in the SN during hypnosis appear to correspond to profound shifts in awareness of internal and external events [123,124]. Additionally, the SN may play a coordinating role in the observed alterations in the dynamics of the ECN and DMN often detected during hypnosis [9,73,74]. Functional changes in the SN during hypnosis are closely associated with changes in awareness of both internal and external events [123,124].

Very recently, Vázquez and co-workers [97] (Table 1) conducted a fMRI-hypnosis study devoted to highlighting neural networks as signatures of hypnosis by contrasting fMRI activity of the common resting state with resting state following hypnotic induction in the absence of goal suggestions (neutral hypnosis). Significant differences in functional connectivity were tested for the following resting state networks: the DMN, ECN, SN, DAN, and SMN. The hypnotic condition disclosed significantly higher functional connectivities than the resting condition for the DAN, SN, and SMN. Still, no significant differences were found for the ECN and the DMN networks. The DAN showed significant connectivity differences in parietal regions not seen in the dorsolateral pre-frontal areas. The SN (embodied by the ACC, Ins, and mPFC) showed increased connectivity with the occipital lobe, mainly the cuneus (Cu) and the calcarine cortex. The SMN showed increased connectivity with the PCC, the retrosplenial cortex, the lingual gyrus (LG), and the Cu. The absence of variations in the connectivity of the ECN (effortful control) during hypnosis may be due to the inherent involuntary nature of hypnosis [1] and to the fact that neutral hypnosis did not exert effort to regulate participant behavior [125,126]. The finding of increased visual cortex connectivity within the SMN can reflect the steady processing of the significant salient visual information of the SN to maintain proprioceptive, visuospatial, and creative timing events in parietal regions [127]. This connectivity pattern could be the mechanism by which hypnotic suggestions can model conversion symptoms or other neurologic and psychiatric conditions. It may be used in psychotherapy to modulate aversive affective experiences.

Lesions in the superior parietal lobe, encompassing the DAN, are linked to motor pathologies akin to ideomotor apraxia [128], a conversion symptom unrelated to neurological damage. In certain psychoses, such as within the paranoid cognitive structure, there exists an entity influencing an individual’s will, a phenomenon mimicked in hypnosis [129]. Thus, hypnosis phenomena can serve as a model for understanding neurological and psychiatric disorders [130].

Another recent fMRI-hypnosis research by De Matos and colleagues [98] (Table 1) was focused on the study of purely intrinsic hypnosis, i.e., a state of physical and mental relaxation, reached with minimal and neutral suggestions. The authors compared fMRI measures and other physiological activities during two different depth levels of neutral hypnosis and the two matching-control conditions. An unbiased whole-brain analysis technique named multi-voxel-pattern analysis (MVPA) served to identify crucial neural centers located in the parieto-occipital-temporal (POT) areas, occipital cortex (OC), LG, Cu, PreCu, and the occipital pole. Comparisons between the two hypnotic states revealed hypnosis-depth-dependent connectivity changes, particularly notable in the left superior temporal/supramarginal gyri, cuneus, planum temporale, and lingual gyri. The MVPA-based seeds used in the analysis revealed specific increases and decreases in functional connectivity patterns across different brain regions. 

Comparable findings are reported in a recent EEG-based modeling study [131], in which a propofol-induced loss of consciousness involved the known neural networks, the DMN, SN, and ECN. Mainly, this report shows that several cortical networks associated with altered states of consciousness may also contribute to hypnosis. For example, areas around a parieto-occipital cluster known as the “posterior hot zone of consciousness” [131] seem unquestionably and distinctly involved. These regions are believed to modulate a broad range of functions linked with altered conscious states [132,133,134,135,136,137,138]. In the de Matos et al. study [98], the lingual gyrus, embedded in the posterior hot zone, was involved in altered network configurations. This finding was present in all statistical comparisons, i.e., within and between levels of neutral hypnosis. Thus, these recent data are in support of the involvement of lingual gyrus, as reported by Landry et al. in their review [22], and offer new insights into hypnosis-induced alterations in functional connectivity and contribute to our understanding of the neurobiology behind hypnosis changes in consciousness.

In sum, neutral hypnosis relates to reduced DMN activity, likely to reflect decreased self-referential thoughts and increased functional connectivity between the ECN and the SN [22]. This holistic perspective on explaining hypnosis is supported by substantial evidence linking ECN to the cognitive processes that underlie cognitive flexibility [139]. Nevertheless, Landry et al.’s [22] findings did not support this view. Instead, their results showed that hypnosis correlates with activation of the lingual gyrus (a brain region involved in higher-order visual processing and mental imagery). Thus, they failed to confirm their lead hypothesis regarding DMN, SN, and CEN involvement. 

Landry and colleagues [22], to account for the role of the lingual gyrus in hypnosis, offered two potential explanations. Firstly, they propose an “intrinsic component of hypnosis” tied to mental imagery. Secondly, they suggest that lingual gyrus activation might be due to “suggestion-specific effects,” wherein visual suggestions play a role in inducing hypnosis.

De Matos and colleagues [98] observed that the lingual gyrus was part of the central cluster common to light- and deep hypnosis conditions compared to control states. However, only two of the studies de Matos and colleagues reviewed explicitly aimed to induce hypnosis through visualization techniques [9,69]. In contrast, the other studies sought to generate non-visual effects, such as hand paralysis [140,141] or non-specific suggestion-guided hypnosis focused on deep mental relaxation [70]. Additionally, one of these studies [70] primarily investigated cortical pain mechanisms, testing whether hypnosis influenced the perception of pain based on pleasantly warm water or painfully hot water placed on the participant’s hand, and hypnosis did not show significant changes in pain perception.

However, in the de Matos et al. study the emerged specific connectome patterns fits into a more general interpretation regarding neutral hypnosis as (*i*) a holistically altered phenomenon of consciousness and as an altered sense of agency manifested in semi-automatic, effortless and involuntary responses [142]; (*j*) a sense of physical relaxation, reported in the light hypnosis condition (probaly due to observed connectivity changes in the cerebellum and in the thalamus, although the observed effects in the cerebellum might be more due to spatial proximity with the broader parieto-occipital cluster rather than direct physiological alterations) [143,144]; (*k*) a sense of deep physical and mental relaxation reported in deep hypnosis condition (paralleled by modified coupling mechanisms in the cortical somatosensory/sensorimotor integration system) and accompained by perceptions of experienced bodily distortions, either of distinct body areas, or as a dissolution of body boundaries, experiential phenomena similar to drug induced altered states of consciousness [145].

Interestingly, Jiang and colleagues [67] have reported intriguing findings regarding brain activity during resting state after hypnotic induction hypnosis (Table 1). The study disclosed reduced activity in the dorsal anterior cingulate cortex (dACC) and increased connectivity between the DLPFC within the executive control network (ECN) and the insula in the salience network (SN). Conversely, they observed a decreased connectivity between the DLPFC in the ECN and the DMN, including the posterior cingulate cortex (PCC). These findings of reduced activity in the dACC align with those later reported by DeSouza et al. [146], who found that higher GABA concentrations in the ACC are linked to increased hypnotizability. However, in a recent review, De Benedittis [147] suggests that the role of the DLPFC in hypnosis may depend on various factors, including the type of suggestion given, which could explain the observed variations in dACC activity.

While there is a consensus regarding decreased DMN functional connectivity during hypnosis, changes in connectivity patterns are debated. Evidence suggests that hypnosis reduces activity in the dACC, increases functional connectivity between the DLPFC (in the ECN) and the insula (in the SN), and reduces connectivity between the DLPFC (in the ECN) and the PCC (in the DMN) [67]. Those reported by Hoeft et al. [148] align with Jiang et al. findings [67], indicating that HHs exhibited more pronounced functional connectivity between the left DLPFC and the SN.

In contrast, Demertzi et al. [74] (Table 1) found reduced connectivity in the posterior midline and parahippocampal structures of the DMN and increased connectivity in the lateral parietal and middle frontal areas during hypnosis. Additionally, in a study evaluating pain perception under hypnosis, Demertzi and her team [123] aimed at quantifying the relationship between external and internal awareness when individuals enter a modified subjective state induced by hypnosis. Their findings demonstrated increased self-oriented processing concurrently with a heightened disconnection from the external environment. Furthermore, brain components responsible for self-awareness and external awareness were less negatively correlated during hypnosis than when individuals were at rest [123]. In a broader sense, hypnosis appears to alter resting-state fMRI networks by diminishing “extrinsic” lateral frontoparietal cortical connectivity, which might reflect a decrease in sensory awareness. In this altered state, the default mode network (DMN) exhibits increased connectivity in its lateral parietal and middle frontal regions but reduced connectivity in its posterior midline and parahippocampal structures [74]. These findings appear in line with those recently reported by Vázquez and colleagues of increased connectivity for the frontoparietal cortices and in the parietal and occipital regions during neutral hypnosis [97], and both authors suggest indicate increased self-oriented processing concurrently with a heightened disconnection from the external environment. The increased functional connectivity found in the parietal lobe may suggest how neutral hypnosis facilitated spontaneous vivid imagery, spontaneous associations, and novel insights since salient visual information was clearly being processed and maintained in the SN and the SMN [149]. This interpretation can help in understanding the underlaying mechanisms of hypnotherapy in terms of modulation of spontaneous visual absorption experiences. It is of great importance to mention here, that, dissociation together with absorption and suggestibility, conforms to what is hypnosis [150,151]. 

However, it is worth noting that other studies have reported contradictory findings. Specifically, Lipari and colleagues [95] (Table 1) observed enhanced activity in the posterior regions of the DMN and decreased metabolic activity in the anterior DMN areas. The authors also observed DMN and complex neural network changes in non-DMN regions, including prefrontal, precentral, cingulate, parietal, and occipital areas. These findings were confirmed by a study by Oakley and Halligan [4]. According to these authors, hypnosis is associated with significant modulation of activity and connectivity that is not limited to the DMN, depending on the depth of hypnosis, the type of mental content, and emotional involvement.

For a critical discussion on divergent study findings regarding the connectivity patterns among regions of interest, see the review by Landry and colleagues [5,22]. 

Despite the heterogeneity among these investigations, the studies mentioned above have produced several consistent results that support participants’ subjective experiences and implicate brain regions and networks associated with those phenomena. Thus, hypnosis has begun to attract renewed enthusiasm from cognitive and social neuroscientists interested in using hypnosis and hypnotic suggestion as models to test predictions about normal cognitive functioning as well as to evoke phenomena such as, e.g., delusional control or hallucinations also present in some neurological and psychiatric disorders [152]. This interest aids in further characterizing hypnosis, thereby bolstering confidence among clinicians to enhance their expertise and more extensively apply hypnotherapy in psychotherapeutic settings for the benefit of patients. However, it is crucial to acknowledge the challenges of generalizing these results due to the diverse imaging techniques employed and the wide-ranging applications of hypnosis in different contexts. It remains an important task to extend these research findings to construct theories of hypnotic suggestion or to utilize this research for highlighting the neural mechanisms underlying responses to suggestion.

Consequently, a challenging issue for research is to disentangle neurophysiological patterns specific to the induction process from those associated with the actual suggestion. This separation is essential since findings indicate changes in the anterior cingulate cortex after induction alone [67] and in response to various suggestions [5,22,153]. Many challenges can be overcome by standardizing induction procedures and suggestion formulations across studies and using a standardized neuroimaging method to investigate brain activity in and out of hypnosis. 

## 4. Structural and Functional Neuroimaging Correlates of Hypnotizability

Although research methods on individual differences in hypnotic susceptibility or hypnotizability have mainly been criticized [28], neuroscientific explorations of hypnotizability offer crucial evidence for constructing a reliable science of hypnosis [154]. The HH individuals are typically identified by their ability to shape their behavior and subjective experience in line with hypnotic suggestions [155]. Not only do they respond more strongly to suggestions, but they also encounter a broader range of unusual experiences during hypnosis compared to LHs [156,157]. 

Several studies have investigated the neural correlates of hypnotic susceptibility, focusing on comparing HH and LH individuals (Table 2). Structural neuroimaging testing reveals anatomical differences in frontal sites, including a larger anterior corpus callosum for HHs, facilitating inter-hemispheric coordination [96,158,159] (Table 2). These volumetric disparities might be reflected in distinct patterns of frontal neural activity among HHs [160].

More recently, as mentioned above, DeSouza and colleagues [146] (Table 2) reported an MRI study investigating individual differences in hypnotizability wherein GABA concentration within the ACC was positively associated with hypnotic induction profile hypnotizability scores. Additionally, an exploratory analysis of questionnaire subscales revealed a negative relationship between glutamate and absorption and individual tendency to imaginative involvement. These results provide a potential neurobiological basis for individual differences in hypnotizability, a result that is useful to guide clinical treatment through hypnosis.

Functional neuroimaging studies partially support the top-down view of hypnosis, indicating distinct cross-network interactions involving the ECN and SN networks in HHs compared to LHs [148,158,164]. Baseline differences in attention-related networks may reflect the higher cognitive regulation capabilities observed in HHs during hypnosis [12]. Further brain imaging findings have reported higher dACC amplitude during rest in HHs than LHs and significantly lower dACC activity in emotional and memory experiences during hypnosis compared to the resting state. Increased functional connectivity between the DLPFC (region of the ECN) and the insula in the SN and decoupling of the DLPFC (part of ECN) and the PCC (part of DMN) have been found during hypnosis [67] (Table 2).

Neurophysiological investigations reported distinct frontal oscillatory patterns linked to hypnotic susceptibility, suggesting that structural neural differences underlie functional activity differences [6]. However, one study challenges this interpretation, demonstrating increased functional connectivity without corresponding structural differences among HHs [148]. In a neuropsychological review, Kihlstrom and colleagues [165] found limited evidence that frontal dysfunction increases hypnotic susceptibility. Nevertheless, the findings suggest that structural and functional differences in the frontal brain contribute partially to hypnotic susceptibility.

Several of the original research suggests that the level of susceptibility to hypnosis primarily relates to variations in top-down regulation. The HH individuals are believed to possess a unique ability to exercise control—via both inhibition and facilitation—over cognitive functions [166,167,168,169]. Numerous studies have explored baseline attention abilities across the spectrum of susceptibility to investigate this hypothesis. However, behavioral findings have been inconsistent (e.g., [64,169,170,171,172,173,174,175,176,177]. Neuroimaging testings report significant individual differences in attention-related baseline activity between HHs and LHs. However, these neural differences do not necessarily correlate with improved task performance but reflect variations in information processing and cognitive style [163,164] (Table 2). 

In line with these observations are those detected in a study comparing analgesic placebo responses that highlighted distinct differences in the DLPFC activity between HHs and LHs without observable behavioral differences between the groups [162] (Table 2). These findings indicate how similar behavioral effects between hypnotizability groups can lead to unique neural patterns associated with top-down regulation and processing style. Therefore, rather than approaching hypnotizability through task performance parameters or control capabilities, researchers might make more remarkable progress by investigating variations in attention strategies and cognitive processing methodologies. 

The hypothesis connecting hypnotizability to differences in cognitive processing has roots in the research of the eighties and nineties [178,179,180,181]. Rather than emphasizing enhanced attention, this framework suggests that HH individuals employ effective cognitive strategies to process suggestions, resulting in better hypnotic responding [182]. Studies supporting this notion reveal substantial variations in response styles even within the group of HHs [23,183,184,185]. In these studies, some HHs possess certain preexisting cognitive traits predisposing them to a marked dissociation and imaginative experience during hypnosis [23,183,186]. Although structural and functional neuroimaging studies generally support the correlation between hypnotizability and variations in top-down brain processes, the observed neural differences should likely reflect diverse processing styles rather than a mere superiority in attentional capacity. 

Interestingly, Santarcangelo and colleagues [187], starting on postural control findings showing that HHs exhibited a less strict postural control, conceptualized hypnotizability as an individual trait responsible for relative variability in postural and visuomotor control and suggested hypnotizability as being involved in constructing individual sensorimotor selves. More recent structural MRI findings have provided evidence of reduced GM volume in the cerebellum of HHs than LHs [161] (Table 2) with significant differences in the left lobules IV and V, which are involved in sensorimotor integration, and in lobule VI, which participates in cognitive-emotional control [188]. Additionally, the HHs showed also gray matter volumes smaller than lows in the right inferior temporal gyrus, middle and superior orbitofrontal cortex, parahippocampal gyrus, and supramarginal parietal gyrus, as well as in left gyrus rectus, insula, and middle temporal cortex at the uncorrected level.

The model reserved a leading role of the cerebellum in hypnotic responding and suggested the involuntariness in hypnotic responding to sensorimotor suggestions as a natural effect rather than merely experienced by an HH individual [187,189]. The morphological variations discovered by these authors expand the conventional focus on the cortex’s role in hypnotizability to include cerebellar regions. This inclusion suggests that unique features in the cerebellum might contribute to differences in sensorimotor integration and emotional regulation related to hypnotizability.

This new conceptualization of hypnotizability seems very interesting and deserves further validation and extensions.

Interestingly, research has shown that disrupting the activity of the left DLPFC using repetitive transcranial magnetic stimulation (rTMS) can enhance hypnotic responsiveness [108]. However, it is worth noting that in the first experiment reported by Coltheart et al. [190], which was an exact replication of Dienes and Hutton’s study [108], the authors did not find any significant change in hypnotic responsiveness after applying rTMS to the left DLPFC. However, in a second experiment by the same authors, wherein hypnotic response was measured objectively, they observed increased hypnotic responsiveness with right-sided DLPFC stimulation. Nevertheless, in a more recent study, Faerman and colleagues [191] have provided results in support of previous Dienes and Hutton findings [108]. Using an original personalized and targeted neuroimaging-guided transcranial magnetic stimulation (TMS, with a continuous theta-burst) to the left-DLPC (L-DLPFC), these authors obtained a temporary increase in hypnotizability and subjective experience of hypnosis. In another recent study by Perri and Di Filippo [192], a unilateral inhibitory transcranial direct current stimulation (tDCS) over the left DLPFC was observed to enhance the experience of hypnotizability by 15.4% and altered a few dimensions of consciousness, such as self-awareness and absorption. 

## 5. EEG Oscillations and Their Associations with Hypnotizability and Hypnosis

Early investigations into EEG patterns during hypnosis reported an increased occurrence of occipital alpha waves in HHs compared to LHs [193,194,195,196,197,198]. Later studies have reported increased alpha activity in HHs during hypnosis [199], as well as after the hypnotic induction procedure [200,201,202]. However, findings from later studies failed to detect an increase in alpha activity with hypnosis [165,203]. However, Franz and co-workers [204] re-processed their-own EEG data from a previously published ERP odd-ball study [205] to investigate whether hypnotic suggestions of a visual obstruction would influence the amplitude of ERP components (N1, P2, and P3b amplitudes). The study also provided source reconstruction and spectro-temporal connectivity analysis of the P3b-related frequency oscillations within the conventional frequency bands (i.e., 1−30 Hz). In response to targets, P3b amplitudes and P3b source were significantly reduced in occipital and parietal brain areas (related to categorization and attention to stimuli).

Additionally, both frontal and parietal electrodes were significantly reduced in effective connectivity within the sole P3-related alpha frequencies (10.5–12 Hz) during hypnosis compared to the control condition. These results provide preliminary evidence that hypnotic suggestions of a visual blockade are associated with a disruption of the coupling within the frontoparietal network implicated in top-down control. Importantly, the effects of hypnosis in terms of P3b amplitudes and effective connectivity did not significantly differ between high, medium, and low hypnotizable participants.

The most consistent relationship between EEG activity and hypnosis is reported in the theta band (4–8 Hz) (for a more detailed review, see [32,33,91]). Several studies have reported increased spectral power in the EEG-theta band during hypnosis [200,203,206,207,208]. Crawford and collaborators [209] said that HHs, compared to LHs, had significantly more significant hemispheric asymmetries (right greater than left) in the parietal region for all frequency bands usually associated with sustained attentional processing, in high-theta (5.5–7.45 Hz), high-alpha (11.5–13.45 Hz), and beta activity (16.5 and 25 Hz). Later, De Pascalis et al. [207] found that only HHs exhibited a higher low-band theta (4–6 Hz) amplitude in bilateral frontal and right posterior areas and a smaller alpha (8.25–10 Hz) amplitude bilaterally in the frontal cortex. Other observations of increased theta activity, particularly in HH individuals, during hypnotic inductions and suggestions [203,210] have been reported. Additionally, HHs tend to exhibit higher baseline theta activity than LHs during hypnosis and waking conditions [203,208,211,212].

There is also evidence of increased gamma activity (above 35 Hz) under hypnosis [213,214]. In an early study, Ulett and colleagues [198] measured a decrease in theta and an increase in alpha, beta, and gamma (40 Hz) activity in the right occipital cortex during hypnotic induction. Further, De Pascalis and colleagues [215,216] reported that HHs exhibited greater 40-Hz EEG amplitude density during emotional states than LHs in nonhypnotic and hypnotic conditions. However, these results were not confirmed by Crawford and coworkers’ [209] observations, although these authors reported a significant beta power increase in the right parietal region. Furter, Schnyer, and Allen [217] found enhanced relative gamma band power (36–44 Hz) in HH participants experiencing recognition amnesia, suggesting that gamma power is associated with hypnotic amnesia phenomena. Isotani and colleagues [218] applied low-resolution electromagnetic tomography analysis (LORETA) tools (first version) for seven frequency bands (full band from 1.5 Hz to 30 Hz) in HHs and LHs during a 2 min eyes-closed resting EEG-recording preceding a hypnotic induction. The authors reported that Fast Fourier Transform (FFT) Dipole Approximation analysis had a significantly more posterior and more left source gravity center for theta (6.5–8 Hz) in HHs, whereas in these subjects, beta-1 (12.5–18 Hz) and beta-2 (18.5–21 Hz) frequencies were localized more posteriorly; LORETA source localization method in LHs showed a cortical anteriorization of beta-1 and beta-2. Finally, Global Dimensional Complexity in the whole band was higher in HHs. Thus, Isotani and colleagues suggested that in a hypnotic context, before hypnosis is induced, the HHs and LHs can be in different brain electric states with the most pronounced posterior brain activations in HHs, whereas anteriorization of brain activation patterns in LHs.

Notably, the most pronounced differences in EEG patterns related to hypnotizability have been reported in the theta spectrum. However, these differences are primarily observed between HHs and LHs, excluding medium hypnotizable individuals (MHs). Crawford [219] proposed a dynamic neuropsychophysiological model of hypnosis involving the activation of the frontal-limbic attentional system. This model posits that attentional and disattentional processes are crucial in experiencing hypnosis, with low theta (3–6 Hz) and high theta (6–8 Hz) rhythms linked to these processes. Accordingly, HHs are thought to possess superior attentional filtering abilities compared to LH individuals, and these differences are reflected in underlying brain dynamics.

Sabourin et al. [203] observed that during hypnosis, both LHs and HHs increased mean theta power (4–7.75 Hz), indicating intensified attentional processes and imagery enhancement. Further research consistently found that HHs tend to exhibit more significant slow-wave theta activity than LHs, both at baseline and during hypnosis, and both groups showed an increase in slow-wave activity after hypnotic inductions [202,206]. However, in the study by Graffin et al. [200], the HHs had higher theta power (3.9–8 Hz) than LHs, mainly in frontal and temporal areas during baseline periods before hypnotic induction. In contrast, the induction of hypnosis produced a decrease in theta activity in HHs. It increased in LHs, particularly in parietal and occipital areas, whereas alpha activity increased across all participant sites. 

Despite these observations, recent studies have not found significant power changes in EEG frequency bands during hypnosis, questioning the hypothesis of theta rhythm as a definitive neurophysiological signature of hypnosis (e.g., [220,221]). Terhune and colleagues [127] also reported increased alpha2 (10.5–12 Hz) power during hypnosis but no significant differences in other EEG bands. Even studies comparing HHs to LHs in the waking state have shown mixed results, with some studies reporting higher theta power in HHs [6,200,222] and others noting no theta differences between the groups [168]. 

Williams and Gruzelier [202] reported increases in alpha power (7.5–13.5 Hz) in HHs at posterior regions during the transition from pre-hypnosis to hypnosis conditions, with decreases observed after hypnosis. The reverse pattern was seen in LHs. Schnyer and Allen [217] reported that HHs exhibited a greater density in the 36–44 Hz frequency band during posthypnosis recognition amnesia, indicating their enhanced ability to maintain focused attention outside hypnosis. Later, De Pascalis et al. [207] reported higher 40 Hz EEG amplitudes in HHs during resting hypnosis conditions compared to LHs. Additionally, Croft et al. [223] found that gamma power (32–100 Hz) predicted pain ratings in non-hypnosis control conditions but not during hypnosis and hypnotic analgesia in HHs, suggesting that hypnosis may alter the relationship between gamma power and pain perception. 

In a recent study, my co-workers and I tested the influences of hypnotizability, contextual factors, and EEG alpha on placebo analgesia [12] using multiple regression and mediation analyses. The study reported that in waking conditions, the enhancement of relative left-parietal alpha2 power directly influenced the enhancement in pain reduction and, indirectly, through the mediating positive effect of involuntariness in placebo responding. In contrast, following hypnosis, the enhanced individual left-temporoparietal upper-alpha (“alpha2”) power did not directly influence pain reduction. However, the indirect mediation effect was significant through the increased involuntariness in placebo analgesia responses. Overall, this study suggests that placebo analgesia during waking and hypnosis involves different processes of top-down regulation.

Some research suggests that enhanced theta oscillatory power may be the most significant oscillatory band associated with hypnosis responding and individual differences in hypnotizability [32]. However, further research is needed to establish theta band rhythm as a definitive neural marker of hypnotizability [5]. While previous research has suggested increased alpha activity in HHs compared to LHs, along with alpha enhancements following post-hypnotic procedures [199,200], hypnosis’ effects on alpha activity have been considered inconsistent [166]. Research on the impact of hypnosis on beta, delta, and gamma activity has been few, making it challenging to formulate specific hypotheses regarding hypnosis’s impact on bandwidth activities other than theta [32]. Additionally, gamma power during hypnosis has shown results in both directions, probably dependent on contextual differences and delivered hypnotic suggestions as well as on the fact that the phenomenon of theta-gamma coupling may be at work, with increases or decreases in gamma activity depending upon waxing and waning phases of theta oscillations [43]. However, more recently, hypnosis research is giving significant emphasis on phase synchrony measures in the gamma band since there is experimental evidence that gamma oscillations may play a key role in various aspects of brain function, including information processing, perception, attention, memory, emotion, and pain [224,225,226,227]. The HHs tend to experience more significant pain relief than LHs in response to hypnosis, and the first also showed substantial reductions in somatosensory event-related phase-ordered gamma oscillations to the obstructive hallucination of stimulus perception during hypnosis [224].

Furthermore, gamma oscillations bind information across different brain regions, resulting in a unified perceptual and cognitive experience [228]. Again, some studies have reported that gamma oscillations are particularly prominent during states of heightened consciousness, such as meditation and the experience of lucid dreaming [229]. However, in other studies, gamma power during hypnosis has shown mixed findings (e.g., [156]). 

Hypnosis theories often have emphasized a top-down mental process within frontal networks involving attention, executive control, and cognitive monitoring in which HH individuals can direct their attention more efficiently, leading to flexible control over their attentional focus [230]. Nevertheless, we still do not fully understand the nature of gamma power fluctuations during hypnosis. According to Jensen et al. [214], the different results across these studies are likely influenced by factors such as the specific hypnotic induction used, the cortical regions under investigation, and the methodologies employed to measure gamma activity. In addition, Lynn and colleagues [231] noted methodological limitations in EEG research, which hinder definitive interpretations of findings. The absence of consistent replications may also contribute to inconsistent results, preventing conclusive statements from being made.

However, it is essential to underline that the inconsistency of outcomes in this domain could be attributed to various methodologies and analyses, each with limitations. 

An example is currently provided by Landry and colleagues [232], who analyzed resting EEG recordings before and after hypnotic induction and used multivariate pattern classification and machine learning to unravel the neural dynamic of hypnotizability using as predictors several neurophysiological features. Among their several neurodynamical findings showing the complex nature of hypnosis, this study provides a novel discovery in the field that the slope of aperiodic non-oscillatory component of EEG spectra is the best predictor of hypnotizability, being significantly greater in HHs than LHs. These findings provide novel evidence that hypnotizability’s predominant discriminative neurophysiological feature is non-oscillatory and promotes the idea that the primary neural distinction in hypnotizability is evident at baseline, even before hypnosis. Indeed, this novel finding aligns with current evidence from the broader field of EEG research, suggesting that aperiodic activity reflects a range of psychologically relevant neurobiological processes and cannot be dismissed as noise. Interestingly, previous research has proposed that the steepness of aperiodic power diminishes across different frequencies (known as the aperiodic slope exponent), which could mirror the balance between neural inhibition and excitation [233]. This notion gains support from studies showing that steeper aperiodic slopes were linked to younger age, faster, more accurate performance in working memory tasks [234], changes in selective attention, and from several other studies suggesting that aperiodic activity could act as a trait-like measure [38,235]. Thus, disentangling aperiodic and periodic components from resting EEG recordings provides novel aperiodic measures to be explored in the field of neuroscience of hypnosis and hypnotizability and allows researchers to revise original hypnosis/hypnotizability neurophysiological results for more reliable findings [236].

## 6. EEG Connectivity of Hypnosis and Hypnotizability

### 6.1. EEG Functional Connectivity Measure

The brain’s electrical activity results from dynamic interactions among distributed neural networks, displaying transient and quasi-stationary processes. The statistical dependency of physiological time series recorded from different brain areas, known as “functional connectivity”, encompasses synchronous oscillatory activity crucial for neural coordination across various cell assemblies involved in multiple systems and representations [237,238]. It is widely accepted that neural synchrony is pivotal in integrating information essential for perception, cognition, emotion, and the representation of consciousness aspects such as body ownership, self-consciousness, and identity [239,240]. One prevalent assumption in hypnosis-suggestion research is that induced alterations in experiential content are associated with distinct changes in functional connectivity. EEG functional connectivity and fMRI research have provided consistent findings showing that HH individuals exhibit distinctive patterns of neural network connectivity.

Measures of dependence between multiple time series, whether linear (coherence, COH) or nonlinear (phase synchronization), can be expressed as combinations of lagged and instantaneous dependence. These measures, with values ranging from 0 to 1, indicate independence when they are zero. They are defined in the frequency domain and apply to stationary and non-stationary time series. These measures find utility in various fields, including neurophysiology, where they assess the connectivity of electric neuronal activity across different brain regions. However, it is essential to note that any measure of dependence in this context may include non-physiological contributions from volume conduction and limited spatial resolution [241]. A conventional measure of functional connectivity between two cortical regions is EEG coherence, believed to reflect the strength of interconnections between cortical areas [242]. EEG coherence between pairs of scalp locations offers valuable insights into brain states, indicating the competition between functional segregation and integration in brain dynamics. However, scalp-recorded EEG coherence has limitations, as even focal brain activity generates widespread EEG voltage patterns.

Conversely, functional connectivity measures based on fMRI and PET have limited temporal resolution and provide only indirect measures of cortical oscillatory activity. Despite its relatively poor temporal resolution, EEG coherence offers a measure of phase synchrony between two-time series, indicating the degree of functional connectivity within the cortex. However, classic EEG-derived functional connectivity measures suffer from volume conduction issues, where spatially separate electrode sites may appear functionally connected despite the absence of information flow [243]. Autoregressive models and Granger causality analysis have been suggested to improve temporal resolution and directionality estimation of information flow (for a review [244]) but do not address the volume conduction problem entirely. Consequently, several synchronization measures have been developed to estimate functional connectivity while controlling for volume conduction, including the imaginary component of coherency (iCOH), phase lag index (PLI) [245], weighted Phase Lag Index (wPLI) [53,246], and source space analysis including LORETA functional connectivity measures of phase synchronization [241,247]. All the PLIs mentioned above are considered robust estimates of the effective phase coupling between two signals because these indexes ignore phase lags of zero since the instantaneous couplings reflect the effects of volume conduction rather than any accurate coupling. These indexes vary between 0 and 1, indicating the extent to which two signals have a phase coupling, with higher values indicating stronger coupling between two brain region signals.

### 6.2. EEG Functional Connectivity Findings under Resting Hypnosis

In a single case study by Fingelkurts and co-workers [248] (Table 3) using a sophisticated EEG functional connectivity measure (Index of Structural Synchrony), the study provided evidence of local and long-distance functional connectivity changes within the cortex after a neutral hypnotic induction and disruption in the functional synchronization among neural assemblies in the left frontal cortex and these changes remained stable even a year later. 

Cardeña and co-workers [156] (Table 3) recorded resting EEG during hypnosis by adopting a neurophenomenological approach to investigate neutral hypnosis (involving no specific suggestion other than to go into hypnosis). They found that hypnotic depth correlated positively with spectral power and power heterogeneity (i.e., the raw power of the Global Field Power curve) for the fast EEG frequencies of beta2 (18.5–21.0 Hz), beta3 (21.5–30 Hz), and gamma (35–44 Hz), but only among HHs, a finding seen consistent with the position reported in previous studies that hypnosis is associated with increased 40 Hz gamma activity, particularly among HHs (e.g., [249]). Following the neutral hypnotic induction, they observed that HHs had spontaneous imagery, positive affect, and anomalous perceptual states that were also associated with lower global functional connectivity during hypnosis. Imagery correlated positively with gamma power heterogeneity and negatively with alpha1 power heterogeneity. Generally, the HHs displayed an opposite pattern of correlations to that found for the Lows. 

**Table 3 brainsci-14-00115-t003:** Experiments that investigated EEG Functional Connectivity findings of resting state following hypnotic induction.

Study	Sample	Experimental Task	EEG Measure	Main Results
Fingelkurts et al. (2007) [248]	A single HH woman, hypnotic virtuoso	Resting state following hypnotic induction	Functional Connectivity	Hypnosis: (1) Lower EEG functional connectivity (Index of Structural Synchrony) for delta (1–3 Hz), alpha (7–13 Hz), beta (15–25 Hz), and gamma (35–45 Hz) frequency bands (except for theta band, 4–6 Hz); (2) Stable functional connection between the right occipital and left inferior-temporal cortex with the highest number of connections for beta, and the lowest for gamma band
Cardeña et al. (2013) [156]	HHs (N = 12), MHs (N = 13) and LHs (N = 12)	Baseline and Seven Resting state following neutral hypnotic induction with hypnotic depth reports	Global Functional Connectivity	Hypnosis: Spontaneous imagery, positive affect, and anomalous perceptual states were associated with lower global functional connectivity in HHs. An opposite pattern of correlations was found in LHs
Li et al. (2017) [250]	HHs, MHs, LHs (SHSS: M = 7.5, SD = 2.8; N = 42 male smokers)	Resting state in Baseline and following hypnotic induction	Coherence	Hypnosis: Increased EEG-delta and theta coherence and reduced alpha and beta coherence during resting hypnosis, suggesting that hypnotic induction yields alterations in consciousness. Higher EEG-delta coherence between specific brain regions predicted reductions in cigarette cravings
Panda et al. (2023) [251]	HHs (N = 9, 6 women)	Resting state following neutral hypnotic induction	Functional Connectivity (weighted phase-lag-index) and Graph Theory Analysis (network segregation and integration)	Hypnosis: (1) Decreased midline and frontal-midline functional connectivities in the alpha (8–11.75 Hz) and beta2 (20–29.75 Hz) bands that were paralleled by a reduction in external awareness and sense of dissociation from the surrounding environment; (2) Increased delta (1–3.75 Hz) band connectivity in frontal and frontoparietal regions, reflecting a heightened state of dissociation; (3) Increased network segregation (short-range connections) in delta and alpha bands, and increased integration (long-range connections) in beta-2 band. These observations may reflect a more effective cognitive processing and a reduced tendency for mind-wandering during hypnosis.
Landry et al. (2023) [232]	HHs and LHs (N = 40, 27 women)	Waking Resting-state and resting state following hypnotic induction	Aperiodic and Periodic Power spectra; Functional Connectivity and Graph Theory Analysis (network segregation and integration)	After the Hypnotic Induction: (1) Opposite patterns of alpha-band and beta-band clustering coefficients, with decreased alpha-band clustering coefficients and increased for the beta-band clustering coefficients. These changes were more pronounced in HHs compared to LHs; (2) Increased global efficiency for theta frequencies and decreased modularity for delta frequencies in HHs compared to LHs

Abbreviations: HHs, high hypnotizable subjects; LHs, low hypnotizable subjects; EEG, electroencephalogram.

Li and collaborators in a resting state EEG study [250] (Table 3) observed altered resting state EEG coherence in individuals undergoing hypnosis treatment for nicotine addiction. The EEG was recorded during two eyes-closed resting periods, one during a waking baseline and the other after hypnotic induction. Hypnosis led to increases in delta and theta coherence and reduced alpha and beta coherence, suggesting that hypnotic induction yields alterations in consciousness. Interestingly, this study’s delta coherence between specific brain regions predicted reductions in cigarette cravings [250]. 

More recently, Panda and colleagues [251] conducted a study comparing neutral hypnosis to an eyes-closed waking-rest condition in HH individuals (Table 3). They used the weighted phase lag index (wPLI) to assess functional connectivity between brain regions and employed graph theory analysis to examine brain network topology in terms of both segregation and integration. The findings of the study revealed several key insights. During hypnosis, the authors observed decreased brain connectivity in the alpha band (8–11.75 Hz) and beta2 band (20–29.75 Hz), particularly in the midline and frontal-midline regions. This reduced connectivity was associated with a reduction in external awareness and a sense of dissociation from the surrounding environment. In addition, they reported an increase in delta band (1–3.75 Hz) connectivity in frontal and frontoparietal regions during hypnosis, seen as reflecting a heightened state of dissociation. Finally, the authors observed, bilaterally in frontal and right parietal electrodes, increased short-range network segregation connectivity in delta and alpha bands and increased long-range integration connectivity in the beta2 band. These modified connectivity patterns and increased network integration–segregation properties were suggested to mirror the modification of internal and external awareness for more effective cognitive processing and a reduced tendency for mind-wandering during hypnosis.

As reported above, Landry and collaborators [232] (Table 3) have conducted an innovative study that extensively explores multiple neural characteristics associated with hypnotizability and hypnotic induction, which are crucial factors in understanding hypnotic phenomena. The study employed multivariate statistics and machine learning to probe the neural dynamics underlying inter-individual differences in hypnotizability. Linear classifiers proved effective in distinguishing HH and LH individuals using neural features from resting-state EEG recorded during pre- and post-hypnotic neutral induction. The authors measured both aperiodic and periodic components of the power spectrum and graph theoretical measures indicating network segregation and integration across all broad frequency bands derived from functional connectivity (i.e., Clustering Coefficient: a measure of the extent of node clustering, Global Efficiency: the level of shared information across the entire network, Global Modularity: the degree to which networks are divided into communities). Regarding the effects induced by neutral hypnotic induction compared to resting pre-hypnosis, linear classifiers detected significant decreases in alpha-band clustering coefficients and increases in beta-band clustering coefficients that were more pronounced in the HHs than LHs. Moreover, hypnosis produced increases in global efficiency within the theta frequencies and decreases in global modularity within the delta frequencies that were more pronounced in the HHs compared to LHs. Interestingly, after controlling for the aperiodic component for the EEG signal, these results reveal that frontal delta power was reduced after hypnotic induction in HHs. 

These findings appear consistent with those reported by Panda and colleagues [251], which underscore how the induction process significantly impacts various facets of neural network dynamics. These results offer proof that the hypnotic induction procedure leads to more pronounced alterations in clustering coefficients among HH individuals in contrast to LH ones.

In sum, common to the reviewed EEG connectivity studies are the findings that the modulation of functional connectivity within and between the frontal and parieto-occipital lobes is essential to account for the phenomenology of neutral hypnosis, with hypnotic induction yielding increases in delta and theta connectivity and reduced alpha and beta connectivity within frontal lobes and between fronto-parietal regions associated with alterations in consciousness.

### 6.3. EEG Functional Connectivity Correlates with Hypnotizability

Gruzelier [167] has proposed a working neurophysiological model of hypnosis and hypnotizability in which HHs under hypnosis are characterized by a reduced upper alpha band coherence between the left frontal and medial electrode pairs. In contrast, increased upper alpha coherence between the same electrode pairs marks the LHs, and decreased coherence within left frontal brain activity during hypnosis delineates the HHs.

Advanced research by Egner and collaborators [64] (Table 4) used a combination of event-related fMRI and EEG coherence recordings with the Stroop task to test predictions from the dissociation theory that hypnosis dissociates executive control and monitoring processes [82,252]. These authors evaluated neural activity in the Stroop task after hypnotic induction without task-specific suggestions. The fMRI results revealed that conflict-related ACC activity interacted with hypnosis and hypnotizability since HHs displayed increased conflict-related neural activity in the ACC during hypnosis compared to baseline and LHs. Interestingly, activity in the DLPFC, associated with cognitive control, did not differ between hypnotizability groups and conditions. HHs exhibited a decrease in EEG gamma band coherence, from baseline to hypnosis, between frontal midline and left lateral scalp sites, while LHs showed a gamma band coherence increase. These findings indicate a decoupling between conflict monitoring and control processes during hypnosis and suggest a negative link between hypnotizability and efficiency of the frontal attention system. These findings align with the view that HH individuals are particularly prone to focusing attention at baseline. However, after hypnosis, their attention control ability appears weakened due to a disconnection between the conflict monitoring and cognitive control processes of the frontal lobe became disconnected.

The study mentioned above findings by Egner and colleagues [64] fit well with the general model proposed by Jamieson and Woody [109], in which breakdowns in the functional integration between different components of executive control networks account for core features in the phenomenology of the hypnotic condition. Alternatively, these findings can also be explained as dissociation or shifts in subjective evaluations of one’s agency during the hypnotic experience, as proposed by the Cold Control theory of hypnosis [83]. The Cold Control theory suggests that hypnosis primarily involves a change in metacognition, where hypnotic responses occur due to a lack of awareness regarding the intentions that drive cognitive or sensorimotor actions, i.e., the activation of sensorimotor network (SMN, Figure 1). For instance, stiffness in a suggested rigid arm might result from intentionally contracting opposing muscles without being conscious of that intent. Similarly, a visual, auditory, tactile, or taste hallucination arises from imagining content without being aware of the intention behind that imagination. 

Interestingly, in line with Egner and colleagues’ report of a functional dissociation between medial and lateral frontal regions in hypnosis were the findings reported by White and colleagues [221] (Table 4) that evidenced a decreased beta EEG coherence between medial frontal and lateral left prefrontal regions in HHs during virtual-reality hypnosis. At the same time, LHs showed increased coherence in the same areas. Additionally, parallel findings are reported by Fingelkurts and co-workers [248] in their single case study with a virtuoso participant. These findings, on the whole, are in support of the Gruzelier neurophysiological model of hypnosis [167] and suggest a link between hypnotizability and the reduced efficiency of the frontal attention system in the left hemisphere. 

Terhune and colleagues [253] (Table 4) recorded the EEG in HH and LH participants during the eyes-closed resting state in control and hypnosis conditions. Synchronization was assessed using the phase lag index (PLI), a phase synchrony measure that controls for volume conduction artifacts [245]. They observed that HHs, compared to LHs, reliably experienced a more significant dissociation state and a lower frontal–parietal phase synchrony in the alpha2 frequency band (10.5–12 Hz) during hypnosis, indicating that high hypnotizability may be linked to functional disruption of the frontoparietal network. These findings correlate with greater posterior upper alpha power in HHs [202].

The above-reported results may reflect a shift from an anterior to a posterior neurophysiological processing mode, as observed in patients with schizophrenia [256]. Resting state studies on hypnosis have traditionally neglected individual differences in spontaneous perceptual states [157,257], and thus, despite quite convergent results, the interpretation of these results could be speculative. Preliminary research addressed this issue and observed that global functional connectivity following a minimal, hypnotic induction was negatively associated with different dimensions of consciousness, including anomalous perceptual states [156]. Cumulatively, these studies seem to suggest that HH individuals exhibit reduced frontal connectivity, although further research is needed to clarify the oscillatory specificity of these effects. 

Hypnosis resting state research by Jamieson and Burgess [220] (Table 4) showed increased posterior connectivity (iCOH, but not COH) from a pre-hypnosis to hypnosis condition in the theta band (4–7.9 Hz) and decreased anterior connectivity in the beta1 band (13–19.9 Hz), with a focus on a frontocentral and an occipital hub, that was greater in HHs compared to LHs. Interestingly, the hypnotic induction elicited a qualitative shift in the organization of specific control systems within the brain in HH participants. Similar observational changes are reported by Isotani and collaborators [218] in a hypnotic context, eyes-closed resting condition, and by Egner and colleagues [64], which reported reduced frontal EEG-gamma connectivity (COH) during a Stroop task performance following hypnosis induction in HHs. These findings support a negative link between hypnotizability and the efficiency of the frontal attention system with hypnotic conditions involving a functional dissociation between conflict monitoring and cognitive control processes.

In a study on hypnotic amnesia, Jamieson and co-workers [254] (Table 4) proved that amnesia suggestions for faces, presented through an old-new recognition paradigm, in hypnosis directly influence memory performance. These authors calculated EEG-lagged nonlinear connectivity as a measure of functional connectivity, including a correction for volume conduction. The authors tested if changes in topographic patterns of upper-alpha (10–12 Hz) EEG oscillations selectively inhibit the recall of memories during hypnotic amnesia as a mechanism of hypnotic dissociation. Behavioral results for face recognition accuracy showed that the used paradigm elicits failures in recognizing recently presented face stimuli in HH participants in response to a hypnotic amnesia suggestion to forget these stimuli. Most importantly, this study indicated that the inability to recognize old faces in response to the amnesia suggestion, and only this condition, is linked to significant increases in evoked upper alpha (i.e., functional inhibition) in the right superior parietal lobule, which provides top-down control for face recollection. In the same condition, upper alpha lagged functional connectivity uniquely increased between the right superior parietal lobule and other functionally specific regions required for recalling recent faces (right parahippocampal gyrus, right fusiform gyrus, and right middle temporal gyrus). 

These results are in line with previous research findings reported by Gruzelier [258,259], Fingelkurts et al. [248], and Terhune et al. [253], who also observed hypofrontality and inhibition in the left hemisphere in HHs experiencing spontaneously an higher dissociation state during hypnosis and showed lower frontal–parietal phase synchrony in the alpha2 frequency band compared to LHs. In sum, all these findings indicate that, following a hypnotic induction, spontaneous dissociative alterations in awareness and perception among HH individuals may result from disruptions in the functional coordination of the frontal–parietal network.

Overall, findings from the above studies indicate that HH individuals exhibit reduced frontal connectivity across various frequency bands during hypnosis. Further research is required to clarify the specificity of these effects. Some of the mentioned studies suggest that variability in hypnotic suggestibility is associated with inter-individual differences in theta and gamma functional connectivity. 

More recently, Keshmiri et al. [255] (Table 4) recorded EEG in HHs and LHs to quantify in response to hypnotic induction suggestions the differential entropy (i.e., the average amount of variation in information in the frequency band of interest) to assess differences in information content in theta, alpha, and beta frequency bands. Results indicate higher hypnotizability is linked to lower theta, alpha, and beta frequency band variability. Additionally, higher hypnotizability correlates with increased functional connectivity in parietal and parieto-occipital regions for theta and alpha, while beta shows no significant change. These connectivity findings were similar to those obtained using the iCOH measure reported by Jamieson and Burgess [220] between pre-hypnosis and hypnosis. However, they also extended the observed effects from neutral hypnosis to hypnotic suggestions.

Furthermore, in terms of differential entropy changes to hypnotic suggestions, findings from this study are in line with several previous findings suggesting the role of the theta band in the transfer of information between the hippocampus and the neocortex [260,261], and alpha activity in reflecting the intensification of attentional processes [262] and transfer of data between functionally connected brain regions. These findings pointed to the engagement of the executive attentional network [11,263] during hypnotic experiences. This view found further evidence in the requirement of the attentional processes for selective enhancement of target-stimulus processing, as well as inhibition of competing processes and responses [67,264,265,266]. 

In their novel EEG study, Landry and colleagues [232] (Table 4) identified the aperiodic exponent of the EEG power spectrum, measured in the pre-induction phase, at the anterior part of the frontal lobe as the key neural feature distinguishing HH and LH individuals (Table 4). Additionally, this neural trait has a frontal topography that parallels prior research, suggesting that responses to hypnotic suggestions are linked to the prefrontal cortex activity [148,164,173,267,268] (Table 2 and Table 4). Regarding functional connectivity findings, the graph theory metrics allowed the authors to evaluate network integration and segregation as a function of hypnotizability. Then, they used a logistic multiple regression analysis to examine the effects of hypnotizability on global efficiency and modularity for all broad frequency bands at the network level. This analysis indicates that global efficiency in delta band activity was a significant positive predictor of hypnotizability. These results suggest that both the exponent of the aperiodic component at the frontal region and global efficiency in the delta band are reliable neural indicators distinguishing individual differences in hypnotizability, supporting prior research in structural and functional neuroimaging [148,158,159] (Table 2). Hypnotizability represents a temperamental trait captured by distinct neural features separate from the hypnotic process. Interestingly, based on the growing body of functional connectivity findings (e.g., as those reported by Panda and colleagues [251]), Landry et al. [232] have suggested that the capacity for hypnotic responsiveness can be most effectively represented by interconnected elements characterized by a structure comprising a core ability that superordinates secondary ones. It could be speculated that the significance of the aperiodic slope in distinguishing between HHs and LHs obtained in the waking-rest state embodies this central component. These novel findings, if validated, will contribute not only to enhancing our understanding of neural processes accounting for hypnosis and individual differences in hypnotizability but will have consequences in the clinical treatments, not only in the therapeutic use of hypnosis but also in considering hypnotizability as a factor significant in the design of a clinical plane. Thus, we need further research to validate the findings mentioned above.

## 7. Conclusions and Future Directions

Resting-state fMRI and structural MRI investigations offer complementary insights into the distinctive frontal functioning associated with high hypnotizability. Reviewed studies indicate that individuals with high hypnotizability display more pronounced reductions in activation of the medial prefrontal or dorsal anterior cingulate cortex following neutral hypnotic induction than their low hypnotizable ones [9,67]. Moreover, research has disclosed that induction-specific reductions in the activation of the DMN regions correspond to spontaneous changes in cognitive and perceptual states, and the state of attentional absorption during a hypnotic induction has been associated with reduced DMN activity and increased prefrontal attentional system activity [73]. Furthermore, fMRI and EEG research using diverse functional connectivity methods has underscored the complexity of neural mechanisms during hypnosis. Specifically, HH individuals, who usually feel the most hypnotized, have shown the highest connectivity between the bilateral DLPFC (i.e., ECN) and ipsilateral insula (i.e., SN) during hypnosis compared to rest. At the same time, for LHs, there are no differences between conditions. This enhanced connectivity during hypnosis is accompanied by reduced connectivity within DMN regions, contributing to altered agency and self-consciousness [67,148]. The intricate relationship between SN, ECN, and DMN further delineates the neural correlates of hypnotizability. Some authors have reported the uncoupling of connectivity between the ECN and the DMN during hypnosis [67].

In contrast, others have found the contrary true [9]. Spiegel and colleagues’ work explained the dissociation between ECN and DMN in response to hypnotic induction as an engagement in the hypnotic state and the associated detachment from internal mental processes such as mind wandering and self-reflection. This explanation reinforces the idea of resting hypnosis as a different state of consciousness rather than a reduced level of arousal [96]. 

According to Demertzi and colleagues’ conceptualization [123], hypnosis alters resting-state fMRI networks by diminishing “extrinsic” lateral frontoparietal cortical connectivity, which might reflect a decreased sensory awareness. In this altered state, the default mode network (DMN) exhibits increased connectivity in its lateral parietal and middle frontal regions but reduced connectivity in its posterior midline and parahippocampal structures [74,123]. However, other studies reported that connectivity in hypnosis is not limited to the DMN, depending on the depth of hypnosis and the ongoing emotional involvement and mental content, making this conceptualization questionable [95,180]. Furthermore, the assessment of neuroimaging assays of hypnosis using the Activation Likelihood Estimation (ALE) method by Landry et al.’s [22] findings did not align with this perspective. Instead, the study demonstrated a stronger correlation between hypnotic responses and activation of the lingual gyrus, suggesting a potential association with mental imagery. Nevertheless, several encouraging research paths point to associations between hypnosis, hypnotizability with neuroimaging functional connectivity and EEG non-oscillatory aperiodic activity that may shed light on individual differences in hypnotic suggestibility and the mechanisms of suggestion. For example, DeSouza et al. [146], using structural neuroimaging measures, have reported a significant positive association between γ-aminobutyric acid (GABA) concentration in the anterior cingulate cortex (ACC) and hypnotizability [146], as well as an inverse relationship between trait hypnotizability and perseveration (served by executive control and the salience systems) [267]. This novel outcome indicates that hypnotizability is a temperamental trait predisposing individuals to be more or less responsive to suggestions. These observations would be highly beneficial for improving clinical treatments with hypnosis. This review has also documented how noninvasive neuromodulation methods (TMS, tDCS) in humans can increase individual hypnotizability [108,191,192]. Thus, it is expected that future hypnosis research should validate GABA/ACC vs. hypnotizability findings and expand upon these findings by setting up reliable adjunctive procedures to enhance hypnotizability, including noninvasive brain stimulation. This validation could open new therapeutic interventions and offer potential pathways for enhancing hypnotic treatment in clinical populations and as a tool for promoting well-being and improving individual potential. 

Finally, studies investigating resting-state functional connectivity during hypnosis and its correlates of hypnotizability consistently suggest reduced frontal connectivity in HHs compared to LHs during hypnosis, particularly in the alpha and beta bands. Moreover, increased posterior connectivity in the theta band and decreased anterior connectivity in the beta1 band in HHs point towards a qualitative shift in brain organization, possibly indicating hypnosis. 

Notably, recent findings reported by Panda et al. [251] in highly hypnotizable subjects disclosed that decreased brain connectivity in the alpha and beta2 bands during hypnosis correlated with reduced external awareness and heightened dissociation. These changes were associated with increased short-range network segregation connectivity in the delta and alpha bands and increased long-range integration connectivity in the beta2 band, underlining shifts in brain network topology. Interestingly, novel EEG findings by Landry and colleagues [232] have selected, using a data-driven approach, the slope of the aperiodic spectral component in the pre-induction phase as the primary neural feature discriminating individual differences in hypnotizability. This outcome aligns with recent findings by DeSouza et al. [146] and suggests that hypnotizability is a temperamental psychological trait predisposing specific individuals to be more receptive to suggestions. This finding highlights the importance of investigating the aperiodic component of the power spectrum in future studies.

Integrating neuroimaging and neuromodulation techniques in future studies will provide exciting prospects for unraveling the multifaceted landscape of hypnosis and its potential therapeutic applications in cognitive neuroscience. Future studies are expected to be designed to validate the link between the neurophysiological measures and hypnotizability suggested by the literature and, e.g., using a conditional process analysis, the causal/hierarchical influence of these variables on hypnotizability. Within this domain, it is also essential to highlight the relationship between ACC GABA concentration and prefrontal slope of aperiodic EEG activity and frontoparietal connectivity changes measured in the waking-state at rest. Additionally, using noninvasive neuromodulation methods (e.g., TMS) to increase the individual hypnotizability level may help evaluate how these measures can change within the same participant. 

According to Kihlstrom [165], scientific developments in this field will demand more intricate experimental frameworks to delve into the specific queries that hypnosis uniquely addresses—those tied to how consciousness monitors and controls. Hypnosis unveils two critical facets: the splitting of awareness, and the sensation of involuntary action.

## Figures and Tables

**Figure 1 brainsci-14-00115-f001:**
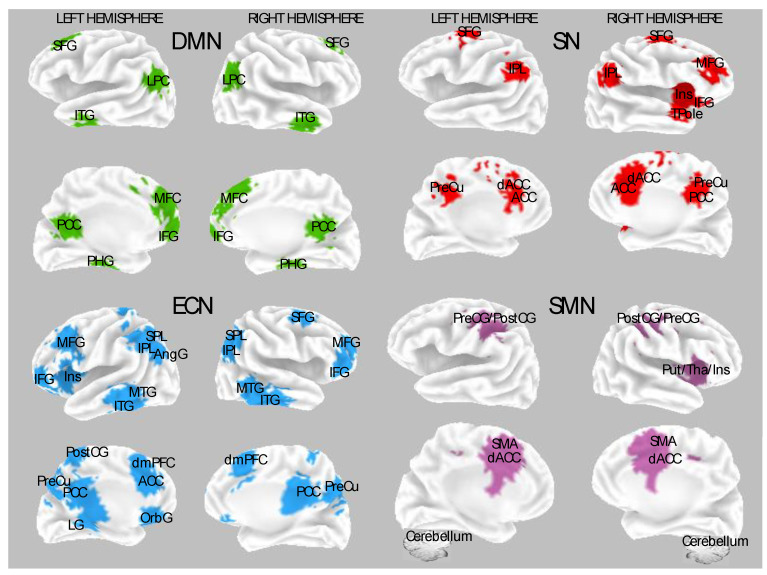
Most reported studies used models to select spatial maps for functional connectivity. The following brain regions constitute four neural networks: default mode network (DMN, upper-left quadrant); salience network (SN, upper-right quadrant); executive control network (ECN, bottom-left quadrant); sensorimotor network (SMN, bottom-right quadrant). Abbreviations: SFG, superior frontal gyrus; LPC, lateral parietal cortex; ITG, inferior temporal gyrus; MTG, middle temporal gyrus; PHG, parahippocampal gyrus; IFG, inferior frontal gyrus; Ins, insula; MFG, middle frontal gyrus; SPL, superior parietal lobule; ACC, anterior cingulate cortex; OrbG, orbital gyrus; PreCu, precuneus; PCC, posterior cingulate cortex; LG, lingual gyrus; SMA, supplementary motor area; dACC, dorsal anterior cingulate cortex; PostCG, post-central gyrus; PreCG, precentral gyrus.

**Table 2 brainsci-14-00115-t002:** Experiments investigating structural and functional imaging correlates of hypnotizability.

Study	Sample	Experimental Task	Technique	Main Results
**Structural Neuroimaging**				
Huber et al. (2014) [158]	LHs and HHs (N = 37 women)	MRI data recording	MRI	HHs: Enhanced grey matter volume in the left Superior and Medial Frontal Gyri and reduced grey matter volume in the left insula. For LHs: Enhanced grey matter volume in Superior, Mid-Temporal, and Mid-Occipital Gyri
Horton et al. (2004) [159]	HHs (N = 8, four men) and LHs (N = 10, five men)	MRI data recording	MRI	HHs: Increased white matter volume in the rostrum of the corpus callosum and more effective attentional and inhibitory capabilities (inhibitory control of pain)
McGeown et al. (2015) [96]	HHs (N = 7), LHs (N = 13) and MHs (N = 9); (N = 29, 17 men)	MRI data recording	MRI	HHs: Greater grey matter volume in the left Mid-Occipital and Mid-Temporal as well as Superior Temporal gyri
DeSouza et al. (2020) [146]	HHs, MHs, and LHs (N = 10 women, N = 10 men)	MRI data recording	MRI	HHs: Higher GABA concentration within the ACC such that the higher the GABA concentration, the more hypnotizable an individual. A negative relationship between glutamate and individual absorption and imaginative involvement tendencies was found
Picerni et al. (2019) [161]	HHs (N = 12, 4 men) and LHs (N = 37, 19 men)	MRI data recording	MRI	HHs: Gray matter volumes in left cerebellar lobules IV/V and VI and in the right inferior temporal gyrus, middle and superior orbitofrontal cortex, parahippocampal gyrus, and supramarginal parietal gyrus, as well as in left gyrus rectus, insula, and middle temporal cortex smaller than LHs.
**Functional l Neuroimaging—Resting Condition**				
Huber et al. (2014) [158]	LHs and HHs (N = 37 women)	Resting-state data recording	fMRI	Reduced connectivity between the Thalamus and the right Fronto-Parietal Network; increase in connectivity between the Posterior Cingulate Cortex and Precuneus with the left Fronto-Parietal Network; enhanced connectivity between the Inferior Parietal Lobule and the Central Executive Network.
Hoeft et al. (2012) [148]	HHs (N = 12, 6 men) and LHs (N = 12, 6 men); (N = 24)	Resting-state data recording	fMRI	HHs: (1) Enhanced functional connectivity between bilateral PCC and Precuneus, and both the lateral visual network and the left frontoparietal network; (2) higher connectivity between the ECN and a right postcentral/parietal area; (3) decreased connectivity between the right frontoparietal network and the right lateral thalamus
**Functional l Neuroimaging—Placebo**				
Huber et al. (2013) [162]	HHs, MHs and LHs; (N = 32, 11 women)	Placebo Analgesia	fMRI	HSs: during the anticipation phase, increased activity in the right DLPC and reduced functional connectivity in the mACC/mPFC (brain regions related to emotional and evaluative pain processing); lower activity in the bilateral anterior thalamus/left caudate regions and left precuneus as well as bilateral posterior temporal foci, during pain perception
**Functional l Neuroimaging - Baseline/Attention Condition**				
Egner et al. (2005) [64]	HHs (N = 11) and LHs (N = 11); (N = 22, 10 women)	Baseline of Stroop Task	fMRI	HHs: Neural responses did not differ between hypnotizability groups for ACC and DLPC
Lifshitz and Raz (2015) [163]	HHs (N = 8, 4 women); LHs (N = 8, 4 women); (N = 16)		fMRI	Enhanced activity in Fusiform Gyrus and Pulvinar
Jiang et al. (2017) [67]	HHs (N = 36) and LHs (N = 21)	Resting, Emotion, Memory	fMRI	HHs: higher dACC amplitude during rest than LHs, and significantly lower dACC activity in emotion/memory conditions during hypnosis compared to rest. Increased functional connectivity between the dorsolateral prefrontal cortex (DLPFC; ECN) and the insula in the SN, and reduced connectivity between the ECN (DLPFC) and the DMN (PCC) during hypnosis.

Abbreviations: HHs, high hypnotizable subjects; LHs, low hypnotizable subjects; MRI, magnetic resonance imaging; fMRI, functional MRI; DLPFC, dorsolateral prefrontal cortex; mPFC, medial prefrontal cortex; ACC, anterior cingulate cortex; dACC, dorsal ACC; PCC, posterior cingulate cortex; DMN, default mode network; SN, salience network; ECN, executive control network; GABA, γ-aminobutyric acid.

**Table 4 brainsci-14-00115-t004:** Experiments that investigated EEG Functional Connectivity correlate of hypnotizability.

Study	Sample	Experimental Task	EEG Measure	Main Results
Egner et al. (2005) [64]	HHs (N = 11) and LHs (N = 11); (N = 22, 10 women)	Baseline of Stroop Task	Coherence	HHs showed a decrease in EEG gamma (30–49.9 Hz) band coherence between frontal midline and left lateral scalp sites after hypnosis, while in LHs, gamma coherence showed an increase
White et al. (2008) [221]	HHs (N = 7) and LHs (N = 10); (N = 17, 9 women)	Resting state following virtual reality hypnotic induction	Coherence	HHs showed decreased beta (13–30 Hz) coherence between medial frontal and lateral left prefrontal sites, while LHs demonstrated an increase in coherence
Terhune et al. (2011) [253]	HHs (N = 28, 21 women) and LHs (N = 19, 13 women)	Resting state following hypnotic induction	Phase Synchrony	HHs reliably experienced a spontaneous greater state dissociation and exhibited lower frontal-parietal phase synchrony in the alpha2 (10.5–12 Hz) frequency band during neutral hypnosis than LHs
Jamieson and Burgess (2014) [220]	HH (N = 12, 2 men) and LH (N = 11, 3 men)	Resting state following hypnotic induction	Imaginary Coherence (iCOH)	Increased theta (4–7.9 Hz) band functional connectivity following hypnotic induction in HHs but not LHs organized around a central-parietal hub. Decreased beta1 beta1 (13–19.9 Hz) iCOH from the pre-hypnosis to hypnosis condition with a focus on a frontocentral and an occipital hub that was greater in high compared to low susceptibles
Jamieson et al. (2017) [254]	HHs (N = 15) and LHs (N = 9); (N = 24, 15 women)	Hypnotic amnesia for face recognition (old-new paradigm)	Lagged-Nonlinear Connectivity	In HHs, the inability to recognize old faces in response to the amnesia suggestion is linked to significant increases in evoked upper alpha (10–12 Hz) and increases in lagged nonlinear connectivity between the right superior parietal lobule, right parahippocampal gyrus, right fusiform gyrus, and right middle temporal gyrus. Synchrony between these regions is suggested as essential for the recall of recent faces
Keshmiri et al. (2020) [255]	HHs (N = 6, 3 women) and LHs (N = 8, 3 women)	Ending phase of hypnotic induction	Differential Entropy; Functional Connectivity	Higher hypnotizability is associated with significantly lower differential entropy (i.e., the average information content) of theta, alpha, and beta frequencies, and this lower variability is paralleled by significantly higher functional connectivity in the parietal and parieto-occipital regions of theta (4–7.9 Hz), and alpha (8–11.9 Hz) frequency bands
Landry et al. (2023) [232]	HHs and LHs (N = 40, 27 women)	Waking Resting State and resting state following hypnotic induction	Aperiodic and Periodic Power Spectra; Graph Theory Measures derived from Functional Connectivity (Clustering Coefficient, Global Efficiency, Global Modularity)	HHs exhibit a greater slope of the aperiodic exponent of the power spectrum across the entire scalp. However, this pattern was particularly pronounced in the anterior part of the frontal site and the right temporal region. The periodic activity did not differ between hypnotizability groups; HHs show greater global efficiency (i.e., the level of shared information across the entire network) in delta band activity during the pre-induction rs-EEG period

Abbreviations: HHs, high hypnotizable subjects; LHs, low hypnotizable subjects; EEG, electroencephalogram; rs-EEG, resting state-EEG; iCOH, imaginary Coherence.

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
