# Peer review of "Brain Functional Correlates of Resting Hypnosis and Hypnotizability: A Review"

_brainsci, 2024, doi:10.3390/brainsci14020115_

Round 1
Reviewer 1 Report
Comments and Suggestions for Authors
Dear Authors
Thanks for writing a good review paper. There are some observations as follows.
1. There are huge misleadings in mentioning abbreviated terms in the whole text. As examples EEG is abbreviated in line11. It is again stated in line 168 which does not follow the standard practice. In line 10 and and 185 PET does not follow the usual and standard practices. The authors are suggested to have a good proof read on the manuscript.
2. Recent references may be added to enhance the quality of the review paper.
Comments on the Quality of English Language
The quality of English is good. But the abbreviated terminologies are not mentioned as per usual practices.
Author Response
Reviewer#1
Thanks for writing a good review paper. There are some observations as follows.
Thank you for your appreciation of my review paper.
- There are huge misleadings in mentioning abbreviated terms in the whole text. As examples EEG is abbreviated in line11. It is again stated in line 168 which does not follow the standard practice. In line 10 and and 185 PET does not follow the usual and standard practices. The authors are suggested to have a good proof read on the manuscript.
- Recent references may be added to enhance the quality of the review paper.
This manuscript's current version has been fully revised and almost all rewritten.
A list of abbreviations is now reported on the front page.
The manuscript has been proofread.

Reviewer 2 Report
Comments and Suggestions for Authors
While the paper offers a comprehensive review of the cognitive neuroscience of hypnosis and variations in hypnotizability, there are several areas where major revisions could enhance its depth, clarity, and potential impact:
1. Clarify the specific scope and objectives of the review. Clearly articulate the gaps or questions in the existing literature that this review aims to address.
2. Provide details on the systematic approach used for selecting and evaluating the literature. Explicitly mention the inclusion and exclusion criteria, databases searched, and the timeframe covered.
3. Expand on the advantages and limitations of functional magnetic resonance imaging (fMRI), positron emission tomography (PET), and electroencephalography (EEG) methods in studying hypnosis. Discuss how each method contributes to the understanding of neural mechanisms during hypnotic states.
4. Provide more in-depth insights into specific findings from fMRI and PET studies, emphasizing regional brain activations and deactivations during hypnosis. Discuss the implications of these findings for understanding the neural correlates of hypnotic phenomena.
5. Elaborate on the nuances of EEG band oscillations as indicators of hypnotic states. Discuss specific frequency bands (e.g., theta, alpha, gamma) and their potential significance in unraveling the neurophysiological basis of hypnosis.
6. Expand on the reviewed functional connectivity findings, offering a more detailed exploration of disruptions in the executive control network. Provide concrete examples and case studies to illustrate how alterations in connectivity may manifest in subjective appraisals of agency during hypnosis.
7. Further explore the integrated functioning of frontal lobes with other cortical regions during hypnosis. Discuss how this integration may contribute to individual differences in hypnotizability, considering both waking and hypnotic states.
8. Offer a more expansive discussion on future research directions. Specify the specific neural networks that warrant further investigation, proposing hypotheses and potential methodologies for advancing our understanding of hypnosis.
9. Strengthen the concluding remarks by summarizing the key findings and highlighting their implications for the broader field of cognitive neuroscience. Discuss how the reviewed literature contributes to resolving existing controversies or uncertainties.
10.Overall Organization: Consider reorganizing the paper to follow a logical flow, potentially organizing sections by the specific neural phenomena discussed. This can aid readers in navigating the extensive information more easily.
11. Consider incorporating visual aids, such as figures or diagrams, to illustrate key concepts and findings. Visual elements can enhance reader comprehension and engagement.
By addressing these aspects, the paper can become a more robust and impactful resource in the field of cognitive neuroscience, providing a comprehensive and nuanced understanding of hypnosis and hypnotizability.
Comments on the Quality of English Language
Minor editing
Author Response
Reviewer#2
While the paper offers a comprehensive review of the cognitive neuroscience of hypnosis and variations in hypnotizability, there are several areas where major revisions could enhance its depth, clarity, and potential impact:
- Clarify the specific scope and objectives of the review. Clearly articulate the gaps or questions in the existing literature that this review aims to address.
The scope and objectives have now been added in the introduction section.
- Provide details on the systematic approach used for selecting and evaluating the literature. Explicitly mention the inclusion and exclusion criteria, databases searched, and the timeframe covered.
Done. It is now reported at the end of the Introduction.
- Expand on the advantages and limitations of functional magnetic resonance imaging (fMRI), positron emission tomography (PET), and electroencephalography (EEG) methods in studying hypnosis. Discuss how each method contributes to the understanding of neural mechanisms during hypnotic states.
Thank you for this suggestion. Paragraph 2 has now been added, including the advantages and limitations of neuroimaging and EEG techniques.
- Provide more in-depth insights into specific findings from fMRI and PET studies, emphasizing regional brain activations and deactivations during hypnosis. Discuss the implications of these findings for understanding the neural correlates of hypnotic phenomena.
Specific findings from fMRI and PET studies are now reported, and activation-deactivation patterns between the main network regions have been reported. An effort to discuss these complex findings is following each study report in paragraphs 3 and 4
- Elaborate on the nuances of EEG band oscillations as indicators of hypnotic states. Discuss specific frequency bands (e.g., theta, alpha, gamma) and their potential significance in unraveling the neurophysiological basis of hypnosis.
It is done in paragraph 5.
- Expand on the reviewed functional connectivity findings, offering a more detailed exploration of disruptions in the executive control network. Provide concrete examples and case studies to illustrate how alterations in connectivity may manifest in subjective appraisals of agency during hypnosis.
I have reviewed functional connectivity findings and expanded by including Tables 1 and 2 for fMRI and PET and Tables 3 and 4 for EEG connectivity studies. The current version focuses more on waking rest and neutral hypnosis findings.
- Further explore the integrated functioning of frontal lobes with other cortical regions during hypnosis. Discuss how this integration may contribute to individual differences in hypnotizability, considering both waking and hypnotic states.
The question of the correlates of hypnotizability is reported in the paragraph 6 (sections 6.2 and mainly 6.3)
- Offer a more expansive discussion on future research directions. Specify the specific neural networks that warrant further investigation, proposing hypotheses and potential methodologies for advancing our understanding of hypnosis.
I have made some effort in this direction. Some specific suggestions are provided in the paragraph 7 (Conclusion and future direction)
- Strengthen the concluding remarks by summarizing the key findings and highlighting their implications for the broader field of cognitive neuroscience. Discuss how the reviewed literature contributes to resolving existing controversies or uncertainties.
I made the concluding remarks by summarizing new key findings (par. 7)
10.Overall Organization: Consider reorganizing the paper to follow a logical flow, potentially organizing sections by the specific neural phenomena discussed. This can aid readers in navigating the extensive information more easily.
I did a great effort, requiring the extension of the deadline, in reorganizing the paper by folling a logical flow.
I hope that with this revised version, I have reached this goal.
- Consider incorporating visual aids, such as figures or diagrams, to illustrate key concepts and findings. Visual elements can enhance reader comprehension and engagement.
Given the short time, I have added only four tables.
By addressing these aspects, the paper can become a more robust and impactful resource in the field of cognitive neuroscience, providing a comprehensive and nuanced understanding of hypnosis and hypnotizability.
Thank you for your patience and help!

Reviewer 3 Report
Comments and Suggestions for Authors
This is a review paper with the title "EEG oscillations and neural functional connectivity underpinning hypnosis and hypnotizability" that focuses on the cognitive neuroscience of hypnosis and variations in hypnotizability by examining research employing fMRI, PET, and EEG methods. My concerns are listed in the following:
1. This manuscript used a lot of acronyms, and it is a little difficult to read smoothly for readers. Sometimes I need to look up the meaning of an acronym from the paragraphs I have already read. I think that it is necessary to add a table about the acronyms, their full names, and their basic usage. It will be more convenient.
2. As a good review paper, it is necessary to summarize the details of all related works and present them in the table format, which can obviously show the properties based on the existing approaches, please review and follow several high-quality review papers in this field. I suggest such tables can be presented at the end of Sections 2, 3, and 4.
3. To facilitate reproducible research, I suggest that the author also mention several public databases from the existing works in this review paper, which would have a positive effect on the academic community.
4. From my understanding, hypnotizability refers to an individual's ability to experience suggested alterations in physiology, sensations, emotions, thoughts, or behavior during hypnosis, meaning it is a subjective behavior influenced by the subjects. Based on that, any related discussions have been included?
5. The review of neuroimaging method is also included in this paper, but the title only emphasizes EEG methods and functional connectivity. The title should be reconsidered in this regard.
Comments on the Quality of English Language
The author should spend time revising this manuscript, the writing still requires further improvement, as there are grammatical errors. Meanwhile, the tense also needs to be improved.
Author Response
Reviewer#3
This is a review paper with the title "EEG oscillations and neural functional connectivity underpinning hypnosis and hypnotizability" that focuses on the cognitive neuroscience of hypnosis and variations in hypnotizability by examining research employing fMRI, PET, and EEG methods. My concerns are listed in the following:
- This manuscript used a lot of acronyms, and it is a little difficult to read smoothly for readers. Sometimes I need to look up the meaning of an acronym from the paragraphs I have already read. I think that it is necessary to add a table about the acronyms, their full names, and their basic usage. It will be more convenient.
A list of abbreviations is now reported on the front page.
- As a good review paper, it is necessary to summarize the details of all related works and present them in the table format, which can obviously show the properties based on the existing approaches, please review and follow several high-quality review papers in this field. I suggest such tables can be presented at the end of Sections 2, 3, and 4.
Thank you. Four tables have been added in the current revised version.
- To facilitate reproducible research, I suggest that the author also mention several public databases from the existing works in this review paper, which would have a positive effect on the academic community.
Public datasets are now reported at the end of the Introduction.
- From my understanding, hypnotizability refers to an individual's ability to experience suggested alterations in physiology, sensations, emotions, thoughts, or behavior during hypnosis, meaning it is a subjective behavior influenced by the subjects. Based on that, any related discussions have been included?
The definition of hypnotizability and some related discussions are now provided in the Introduction.
- The review of neuroimaging method is also included in this paper, but the title only emphasizes EEG methods and functional connectivity. The title should be reconsidered in this regard.
Thank you. The new title is: "Brain Functional Correlates of Resting Hypnosis and Hypnotizability: A Review."
The author should spend time revising this manuscript, the writing still requires further improvement, as there are grammatical errors. Meanwhile, the tense also needs to be improved.
You are right! I spent a lot of time on it and hope I have improved the manuscript to an acceptable form.
Round 2
Reviewer 3 Report
Comments and Suggestions for Authors
The author addressed my queries raised and I am satisfied with the answers. The manuscript has been improved, so I recommend it for acceptance.